



# Modelling the hydrological interactions between a fissured granite aquifer and a valley mire in the Massif Central, France

Arnaud Duranel[1,2], Julian R. Thompson[1], Helene Burningham[1], Philippe Durepaire[3], Stéphane Garambois[4], Robert Wyns[5], Hervé Cubizolle[2]

[1] UCL Department of Geography, University College London, London WC1E 6BT, United Kingdom
[2] Lyon University, UMR 5600 CNRS EVS, 42023 Saint-Etienne cedex 2, France
[3] Conservatoire d'Espaces Naturels de Nouvelle-Aquitaine, Réserve Naturelle Nationale de la Tourbière des Dauges, Sauvagnac, 87340 Saint-Léger-la-Montagne, France
[4] Université Grenoble Alpes, Univ. Savoie Mont Blanc, CNRS, IRD, IFSTTAR, ISTerre, UMR 5275, 38041 Grenoble, France
[5] Bureau de Recherches Géologiques et Minières, ISTO, UMR 7327, 45060 Orléans, France

*Correspondence to*: Arnaud Duranel (arnaud.duranel@univ-st-etienne.fr).

**Abstract.** The contribution of groundwater to the hydrology of hard rock regions has long been assumed to be small. This is being progressively challenged and conceptual hydrological models of headwater wetlands in these regions may need to be revised. We developed a high-resolution MIKE SHE / MIKE 11 model of a 231.3 ha headwater catchment in the granitic uplands of the French Massif Central to estimate the contribution of groundwater upwelling to the water balance of the Dauges mire, an acidic valley mire of international importance for nature conservation. We estimated that groundwater upwelling from the underlying granite weathering formations – mostly an approximately 55m deep fissured zone – provides 27.1 % of total long-term inflows to the mire. This contribution increases to 37.2 % in September when total inflows are small. Overland boundary inflow accounts for an average of 40.2 % of total inflows. However most of this originates from groundwater seepage through mineral soils along the mire margins or in small unchannelized valleys upslope of the mire. A sensitivity analysis showed that model performance in terms of the simulation of mire groundwater levels was most sensitive to parameters describing the mineral soils and granite weathered formations rather than the overlying peat layer. Variation partitioning showed that groundwater upwelling was the most important factor driving simulated monthly groundwater table depth within the mire. Sustained groundwater upwelling maintains the mire water table close to or at ground level for most of the year. As a result, precipitation and overland boundary inflows are mostly evacuated as saturation-excess runoff. There was close agreement between the observed distribution of mire habitats and areas where the simulated long-term groundwater seepage rate was larger than zero in September. Groundwater upwelling from the underlying weathered formations can be a quantitatively important and functionally critical element of the water balance of valley mires in granitic headwater catchments. These results have important legal and management implications.



## 1 Introduction

Wetlands, and in particular mires (i.e. wetlands actively accumulating peat, Rydin and Jeglum, 2006), are widely recognised as providing multiple ecosystem services (Okruszko et al., 2011). Mires are more efficient than any other terrestrial ecosystems at sequestering carbon. Despite occupying only 3 % of the world's land area, they contain twice as much carbon as all forests

(Frolking et al., 2011; Lindsay, 2010; Parish et al., 2008; Worrall et al., 2011; Yu et al., 2011). Mires are characterised by very distinctive environmental conditions and thereby support unique ecosystems and many specialised species (Parish et al., 2008). The long-term provision of these services depends on peat remaining waterlogged, and therefore on the long-term stability of the mire water balance.

Groundwater inflow has long been recognised to have a critical role in the water balance of many mires in sedimentary contexts

(Boeye and Verheyen, 1992; Gilvear et al., 1993; House et al., 2016; Koerselman, 1989; Rossi et al., 2012; Siegel and Glaser, 1987; Wassen et al., 1990). In contrast, the importance of groundwater to the water balance of wetlands in hard rock regions, especially in upland and mountainous areas, has most often been considered, if not negligible, at least less significant. This is based on the assumption, common to both hydrogeologists and hillslope hydrologists, that beneath the soil and saprolite (i.e. the loose in situ products of weathering), fresh bedrock is mostly impermeable, even though discrete water bearing zones

(attributable to fractures of tectonic origin) can be found locally. Hard rock aquifers are often seen as compartmentalised and discontinuous (Lachassagne, 2008). Similarly, most hillslope hydrologists have implicitly assumed that upland hard rock catchments are quasi-impermeable beneath the soils and that shallow subsurface lateral flow through the soil is the main source of stream flow (Banks et al., 2009; Gabrielli et al., 2012; Haria and Shand, 2004; Tromp-van Meerveld et al., 2007). However, in the last two decades, hydrogeologists have progressively realised that classical conceptual models of groundwater flow in

hard rock regions, in particular within granitic areas, may need to be revised. Granite weathering over sufficiently long periods and under suitable environmental conditions can produce a stratiform lateritic profile, which, in addition to the saprolite, includes a densely fissured layer typically 40–50m deep (Dewandel et al., 2006; Lachassagne, 2008). This layer is characterised by numerous sub-horizontal fractures of weathering origin (Lachassagne et al., 2011; Maréchal et al., 2004b), the density and hydraulic connectivity of which decrease with depth until the fresh bedrock (Guihéneuf et al., 2014). The fissured layer has a

relatively high hydraulic conductivity but low porosity, and provides the transmissive function of a continuous stratiform composite aquifer (Maréchal et al., 2004a). In parallel, hillslope hydrologists have questioned the 'impermeable bedrock dogma' (Tromp-van Meerveld et al., 2007) and re-evaluated the role of 'deep' groundwater in streamflow generation in granitic areas. For example, investigations in the Kyriu Experimental Watershed in Japan showed that the contribution of 'deep' groundwater to stream discharge was far from negligible: 65 to 71 % of annual discharge in a 6 ha sub-catchment (Kosugi et

al., 2006), and 50 to 95 % in another 0.1 ha sub-catchment (Uchida et al., 2003). In the latter case, this contribution originated from a seepage area covering only 0.5 to 2.0 % of the total sub-catchment. Groundwater in the fissured zone has also been found to be a highly dynamic component of headwater systems. It is characterised by rapid recharge pathways resulting in fast responses to rainfall events (Gabrielli et al., 2012; Haria and Shand, 2004; Kosugi et al., 2011) and can play an essential role





in the hydrological connectivity between hillslopes and riparian areas, particularly at longer-than-event time-scales (Tromp-van Meerveld et al., 2007).

The contribution of 'deep' groundwater to the water balance of many headwater mires in hard rock regions may, therefore, have been overlooked, in particular where these mires are located in topographic lows. A small number of observational studies

have suggested that such mires correspond to areas of sustained groundwater upwelling and seepage (Branfireun and Roulet, 1998; Morley et al., 2011; Šanda et al., 2014). However, direct long-term quantification of groundwater upwelling at a high spatial resolution through field measurements is currently impractical, and as a result these studies provide limited insights on the degree of groundwater dependence of these systems. Physically based, integrated, spatially distributed modelling approaches provide an alternative way to constrain estimates based on limited data, and are often associated with smaller errors

than estimates based solely on field observations (Gilvear and Bradley, 2009). However, the use of physically based spatially distributed models at high resolution in hard rock regions is complicated by the large contrast in hydrodynamic properties between the largely impermeable matrix and the water-bearing fractures, and by the small-scale variability in the density and connectivity of fractures in the fissured zone (Levison et al., 2014). Integrated hydrological models able to represent groundwater flow in fractured media are still scarce (Brunner and Simmons, 2012), and the hydrodynamic properties of the

fissured zone are extremely difficult to characterise at high resolution (Singhal and Gupta, 2010). As a result the vast majority of modelling studies have used the equivalent porous medium approach, assuming that the fractured medium behaves as a continuous porous medium with more or less homogeneous hydrodynamic properties at larger spatial scales (Long et al., 1982). There is a relative consensus that this approach provides acceptable results at the regional scale, but poorly reproduces local flow systems (Singhal and Gupta, 2010). The applicability of the equivalent porous medium approach to provide high-

resolution estimates of groundwater inflow to small headwater wetlands in hard-rock regions is therefore unclear.

Here we apply the MIKE SHE / MIKE 11 modelling system to an acidic valley mire in the granitic uplands of the French Massif Central. Our objectives are 1) to test the ability of an equivalent porous medium approach with limited data on the hydrodynamic properties of the granite weathering formations to reproduce high-resolution spatial and temporal patterns in groundwater seepage and groundwater table depth within the mire, 2) to quantify the mire water balance including its

dependence on groundwater inflows from granite weathering formations, and 3) to investigate the hydrological processes driving groundwater table depth in the mire.

## 2 Methods

### 2.1 Research site

The 231.3 ha catchment that is modelled broadly matches the boundaries of the Dauges National Nature Reserve (NNR),

located near Saint-Léger-la-Montagne, in the administrative department of Haute-Vienne, Nouvelle-Aquitaine, France (latitude: 46° 00' 42" N, longitude: 1° 25' 07" E, Figure 1). It belongs to the Monts d'Ambazac, a low altitude Hercynian mountain range at the north-western limit of the Massif Central. The catchment lies entirely on two-mica leucogranite,



dissected by numerous veins of lamprophyres. It is a typical granitic etch-basin (Valadas, 1998), and comprises a circus-like valley with a flat bottom surrounded by gentle hills that opens into a narrow linear valley leading to another etch-basin further downstream beyond the research catchment outlet. Elevation ranges from 532 m above sea level (NGF69) at the catchment outlet to 664 m at the top of Puy de la Garde, on the south-eastern boundary of the catchment. A 30 m high residual hill called

Puy Rond rises at the centre of the etch-basin. The site was designated as a NNR in 1998 mainly to ensure the conservation of a range of acidic mire habitats located at the bottom of the etch-basin and covering a total area of 43 ha (Durepaire and Guerbaa, 2008). It was further designated as a Special Area of Conservation (FR7401135) under the EU 92/43/EEC Habitats Directive. Most of the mire habitats have been identified as belonging to "Raised bogs" habitats (as defined in the Corine Biotope classification, Commission of the European Communities, 1991), purely on a floristic basis (Durepaire and Guerbaa, 2008).

However the surface topography of the mire does not resemble that of a raised bog even though a number of microforms may possibly be partly or fully ombrotrophic. Other major habitats include mat-grass swards, acid purple moor grass meadows, mire willow scrub, *Sphagnum* birch woods, acidic fens and transition mires. The mire's catchment is dominated by semi-natural beech (*Fagus sylvatica*), oak (*Quercus robur*) and chestnut (*Castanea sativa*) woodlands, with some permanent acidic grassland and heathland patches scattered across the site. It lies at the transition between an altered oceanic climate and a

mountainous climate (Joly et al., 2010). Long-term (1981–2010) mean annual precipitation and temperature recorded 4.2 km away from the site at an altitude of 629 m are 1367 mm and 10.1 °C respectively. Precipitation is relatively well distributed throughout the year.

**2.2 Geological model**

A conceptual 3D geological model was built based on a range of field investigations including topographic surveying,

geological drilling, electrical resistivity tomography, manual augering and probing, and analysis of existing outcrops. This model is detailed in Duranel (2015) and is only briefly summarised here.

Three different datasets were aggregated to produce a digital elevation model (DEM). Within the mire, surface elevation was measured using a differential geo-positioning system (DGPS) at approximately 5 m resolution and with an accuracy of around 10 cm. On mineral soils in the most southern part of the catchment, topographic data was extracted from the IGN BD Topo, a

nation-wide DEM with a resolution of 25 m and a root mean square error (RMSE) of 4.8 m against DGPS ground-truthing points within the site. Finally, on mineral soils in the rest of the catchment, the DEM was based on mass points and contour lines at 1 m intervals digitised from existing 1:1000 topographic maps, produced using both traditional surveying and stereo-photogrammetry. Checks carried out using DGPS have shown that the accuracy of mass points is within ± 0.1 m.

Large-scale underground uranium mining was undertaken in the area during the second part of the 20[th] century. Logs of seven

geological boreholes drilled in 1973 to depths of between 67 m and 173 m for the purpose of uranium exploration were provided by the mining company Areva (now Orano). These were all drilled along the eastern margin of the mire, across presumed mineralised faults located using ground surface radiological surveys. Electrical resistivity tomography (ERT) was used to estimate the depth of granite weathering formations along three 380 to 868m long transects located further east across





the wetland and the lower hillslopes, and one 315 m long transect located on the southern hilltop. A combination of Schlumberger and Wenner arrays were used, with an electrode spacing of 5m and a maximum penetration depth of 60m below ground (Duranel, 2015). Existing sections cutting through peri-glacial deposits and weathering formations and resulting from small-scale grus extraction or from road construction were also described.

Together, these investigations showed that periglacial deposits are very patchy and can be considered to be hydro-dynamically equivalent to in situ saprolite. ERT only detected in situ saprolite, less than two metres deep, on an intermediate hilltop along the western boundary of the catchment. In situ saprolite has been almost completely eroded from other hilltops and hillslopes. The presence of substantial depths of saprolite underneath the wetland is still uncertain. Geological drilling showed the presence of a 15–40 m deep saprolite layer north-east of Puy Rond, however the boreholes were all drilled within a small area

along uranium-rich mineralised faults and may not be representative of the rest of the basin. Nevertheless they show that saprolite has not been completely eroded from the bottom of the basin above Puy Rond. However, ERT surveys did not provide evidence of saprolite at the bottom of the basin downstream of Puy Rond.

ERT transects identified the presence of a densely fissured granite zone at least 50–60 m deep at all investigated locations, with some variations in fissure density, seemingly larger underneath topographic lows. This is consistent with the geological

drilling logs, which suggested a thickness ranging from 38 m to 65 m (average: 54 m). On all profiles the margins of the mire coincided with the location where the inferred groundwater table reaches the ground surface, suggesting a determining role of groundwater seepage in the mire water balance and a high degree of hydrological connectivity between the mire and the hard-rock aquifer.

Peat deposits were mapped using manual augering and probing. The average peat depth within the mire was 0.80 m (standard

deviation 0.49 m), with a maximum of 3.45 m at its centre (Duranel, 2015). Stratigraphical surveys and slug tests showed a rapid increase in peat humification and a decrease in hydraulic conductivity with depth. Peat hydraulic conductivities measured between 0.1 and 0.2 m below ground level had a median value of $2.6 \times 10^{-5}$ m s$^{-1}$. Values recorded at depths ranging from 0.6 to 1.5 m below ground level were 2–3 orders of magnitude smaller (median value: $4.3 \times 10^{-8}$ m s$^{-1}$ ). These field investigations demonstrated that peat properties were highly variable across the site. Alluvial deposits of sand and small gravel, up to 1.7 m thick, were found beneath the peat along the stream downstream of Puy Rond. Further away from the stream, similar material

was found to form 10 cm to 110 cm thick patchy lenses intermixed with peat deposits.

Pedological pits dug by Verger (1998) and Gratia (in Duranel, 2015) showed that soils outside the mire are generally relatively deep (40–70 cm), even on hilltops and relatively steep slopes such as those found on Puy Rond. According to the French pedological classification (Baize and Girard, 2009), they mostly belong to acidic *podsosols ocriques, allocrisols* and *brunisols*.

Texture is loamy-sandy to sandy-gravelly, and drainage is always good.

## 2.3 Hydrometeorological monitoring

Groundwater level was monitored from December 2010 to October 2013 in a network of 16 shallow dipwells installed within the mire and on the lower mineral slopes (Figure 1). These were equipped with automatic loggers (Mini-Diver, Schlumberger)



during all or part of the monitoring period. The 15 min records were corrected for atmospheric pressure recorded by a barometric logger (Baro-Diver, Schlumberger) installed at the site and for logger drift based on regular manual checks, and aggregated to daily mean values (Duranel, 2015).

Discharge was measured from January 2011 to December 2013 at the outlet of the main mire area (Pont de Pierre, Figure 1)
using an Orpheus-Mini (OTT) level logger. The 15 min stage records were corrected for logger drift based on regular manual checks, and converted to discharge on the basis of a rating curve established from spot discharge measurements undertaken using dilution gauging or an electromagnetic flowmeter and the velocity-area method (Duranel, 2015). Discharge was also measured using sharp-crested V-notch weirs and float loggers (Thalimèdes, OTT) from January 2011 to June 2013 at three locations in the upper reaches of the mire (Girolles, Marzet and Rocher, Figure 1). Stage records were corrected for logger
drift, converted to discharge using the equation provided by Dingman (1994, p. 544) and aggregated to daily mean flow.

Radiation, air temperature, relative humidity, wind speed and rainfall were measured 2 m above ground at the centre of the mire using an Enerco 404 (Cimex) meteorological station from June 2010 to March 2013. Generalised least-square regression models were used to reconstruct long-term (01/08/1998–31/12/2013) daily meteorological time-series based on records from nearby Météo-France permanent weather stations, including a station located 4.2 km from the centre of the research site. The
predicted dataset was used to extend the observed time-series and infill missing records (Duranel, 2015). Reference evapotranspiration (ETo) was calculated using the FAO Penman-Monteith method (Allen et al., 1998).

## 2.4 Model development

The MIKE SHE / MIKE 11 modelling system (Graham and Butts, 2005; Refsgaard et al., 2010) was used to model the mire and its catchment. MIKE SHE / MIKE 11 is a deterministic, fully distributed modelling system able to model actual
evapotranspiration, unsaturated zone flow and storage, saturated zone flow and levels, overland flow and channel flow processes in an integrated manner. It has previously been successfully applied in a wide range of wetland environments (e.g. Al-Khudhairy et al., 1999; Bourgault et al., 2014; Clilverd et al., 2016; Hammersmark et al., 2010; House et al., 2016; Thompson, 2004; Thompson et al., 2004, 2017). Originally fully physics-based, it has evolved to include a suite of modules that also allow for the use of simplified physical equations or even conceptual models based on distributed, semi-distributed
or lumped approaches (Refsgaard et al., 2010).

In the present study, a combination of both physics-based and conceptual modelling approaches was used. A 10 m × 10 m grid was employed throughout the model resulting in 23111 model cells. The empirical two-layer unsaturated zone model (Yan and Smith, 1994) was selected. This was principally to reduce computation time, but was also justified by the lack of detailed data on unsaturated zone properties since the lower number of parameters required by the two-layer model facilitates their
estimation through calibration. Due to the two-layer model's limitations in representing interception in woodland, evaporation from interception was modelled outside MIKE SHE using the same equations as those implemented in the HYLUC model (Calder, 2003), with a distinction being made for tall vegetation (woodland, heath and shrubs) between wet-time evaporation (corresponding to evaporation from interception, modelled outside MIKE SHE) and dry-time evapotranspiration (which





includes transpiration and any other evaporation from the ground surface, simulated using MIKE SHE's two-layer model). Calder's γ parameter was calibrated against the mean of bulk interception ratios (annual for evergreen conifers and heathlands, seasonal for deciduous woodlands) cited in the literature for similar vegetation types and measured in similar climatic conditions (Delgado et al., 2010). The method proposed by Allen et al. (1998) and Allen and Pereira (2009) was used to

estimate the basal crop coefficient $K_{cb}$ (equivalent to Calder's β) based on climate, vegetation height, leaf area index and mean leaf resistance. As recommended by Calder (2003), no distinction was made between wet time evaporation and dry time evapotranspiration for short vegetation (grassland and wetland), and therefore evapotranspiration was fully modelled within MIKE SHE. Crop coefficients for short vegetation types were taken from the literature. The values used for parameterising the evapotranspiration model are shown in Table 1.

Saturated flow was modelled using an iterative implicit finite difference technique to solve the 3-dimensional Darcy equation including within the fissured granite for which an equivalent porous medium approach (Long et al., 1982) was used. A no-flow boundary was assumed around the boundaries of the topographic catchment. At the downstream end of the catchment, a constant-gradient boundary was used, assuming that the slope of the phreatic surface broadly follows that of the topographic surface. The geological model described above was simplified into two saturated zone computational layers. The lower layer

represented the fissured granite, with a constant depth of 55 m below the topographic surface. Within the mire, the bottom of the upper layer followed the boundary between peat deposits and the underlying mineral formations, with a minimum depth of 0.5 m to avoid numerical instabilities. On mineral soils, the upper layer was defined as 2 m deep to match the estimated root depth of woodlands, and therefore corresponded to a complex of soil, peri-glacial formations, in situ saprolite and, in a substantial part of the catchment, the top of the fissured granite layer. To reduce the complexity and running time of the model,

the thick in situ saprolite occurring locally upstream of Puy Rond and the alluvial deposits beneath the peat downstream of Puy Rond were not represented. Similarly, the mire acrotelm could not be differentiated from the catotelm due to its small thickness and the peat layer was represented as a unique layer with homogeneous properties. In the absence of data on the saturated and unsaturated hydraulic properties of local granite weathering formations, they were assumed to be spatially homogeneous. Overland flow was modelled using a 2D, finite-difference, diffusive wave approximation of the Saint-Venant

equations, solved using the successive over-relaxation method.

The MIKE SHE model was coupled to a MIKE 11 1D river model of the main channels within the catchment. In order to achieve logistically feasible run times during calibration and validation, channel flow was modelled using a kinematic routing method, and water levels estimated a posteriori based on Manning's equation. This method is fast, and generally relatively accurate for fast-flowing streams where no backwater occurs (DHI, 2009b). It is not expected to give an accurate representation

of stream stage, and as a consequence it was assumed that exchanges between the overland flow component and watercourses occur towards the latter only (i.e. there is no over-bank flooding). This simplification is acceptable since within the research site over-bank flooding only occurs within a very narrow band along the main stream and in a limited area in the downstream part of the mire.


The Dauges MIKE SHE / MIKE 11 model used an adaptive time-step, whilst results were aggregated to a daily resolution. Model outputs were extracted for the period 01/01/2001–31/12/2013, after an approximately 2.5-year warm-up period beginning on 01/08/1998 (corresponding to the installation of the nearest permanent automatic meteorological station).

### 2.5 Model calibration and validation

The model was manually calibrated and validated against observed discharges and groundwater table depths. To accommodate slightly different data availabilities, the calibration period was set as 01/01/2011–30/06/2012 for discharge and 01/01/2011–31/12/2012 for groundwater table depth. The validation period was set as 01/07/2012–31/12/2013 for discharge and 01/01/2013–31/12/2013 for groundwater table depth. Model performance was evaluated using the Nash-Sutcliffe Efficiency (NSE) and percent bias (PBIAS) statistics recommended by Moriasi et al. (2007), and a visual assessment of the ability of the
model to reproduce seasonal groundwater table patterns. Calibration parameters together with their calibrated values are shown in Table 2.

The ability of the calibrated model to simulate realistic groundwater tables across the catchment was further investigated by validating a raster map of the mean groundwater table depth simulated across the site for the 2001–2013 period against the observed mire boundaries, mapped using a combination of botanical (Durepaire and Guerbaa, 2008) and pedological (Duranel,
2015) criteria. The groundwater table depth threshold best discriminating between mire and other habitats was found by optimising a Cohen's kappa agreement function (Congalton, 1991) between simulated and observed mire boundaries.

### 2.6 Sensitivity analysis

The sensitivity of the calibrated model was evaluated by assessing the rate of change in the root mean square error for simulated discharge and groundwater table depth when a single parameter was perturbed by a defined proportion (Rochester, 2010). The
sensitivity of the model to a parameter $i$ is calculated following Eq. (1):

$$S_i = \frac{\partial RMSE}{\partial \theta_i}(\theta_{i,upper} - \theta_{i,lower}) \tag{1}$$

where $\theta_i$ is the model parameter investigated, and $\theta_{i,upper}$ and $\theta_{i,lower}$ are the user-specified upper and lower limits of the parameter (see Table 2). Scaling by the parameter range allowed for the comparison of local sensitivity coefficients between parameters of different scales of magnitude. The sensitivity was evaluated locally around a specified set of parameters ($\theta_1$, $\theta_2$,
..., $\theta_n$), only one of which ($\theta_i$) was perturbed. A perturbation fraction of 5 % of the range was used, implementing both the backward and forward difference approximation methods (DHI, 2009a). The sensitivity of the model to the depth of the granite fissured zone was also tested although this parameter was not used for calibration.

### 2.7 Water balance analysis

A water balance was computed using the MIKE SHE water balance tool for two areas upstream of the main mire outlet
corresponding to mineral and peat soils, respectively. These water balance domains cover 36.9 ha and 125.0 ha, respectively





(Figure 2). The water balance was not calculated over the entire model domain for two reasons: firstly to reduce potential errors caused by uncertainty in boundary conditions at the north-west end of the model domain as well as the relatively sparse hydrogeological and hydrological data downstream of Pont de Pierre; and secondly to constrain the focus of the analysis on the main designated mire area upstream of Pont de Pierre. The stream flowing through the mire was not included in the water

balance of the mire (i.e. only fluxes between the stream and peat soils or the surface of the mire were considered with flow in and out of the stream where it enters and exits the mire excluded).

Simulated groundwater table depth, upward saturated flow from the lower computational layer (hereafter referred to as groundwater upwelling) and exchange from the saturated zone to overland flow (hereafter referred to as groundwater seepage) were mapped across the entire catchment.

**2.8 Processes driving changes in groundwater table depth in peat soils**

Processes driving changes in groundwater table depth in peat soils were investigated using variation partitioning (Legendre and Legendre, 1998), based on simulated spatially averaged monthly means to focus on time scales relevant to vegetation (Wheeler et al., 2004). Explanatory variables included precipitation, groundwater upwelling, overland boundary inflow, actual evapotranspiration and groundwater table depth in the preceding month. An exploratory analysis showed that the relations

between groundwater table depth and explanatory variables were mostly non-linear, therefore variation partitioning was undertaken in a generalised additive modelling (GAM) framework (Wood, 2016).

**3 Results**

**3.1 Model performance**

Figure 3 and Figure S1 in the Supplement show observed and simulated daily discharge at the four gauging stations during the

calibration and validation periods. According to the guidelines of Moriasi et al. (2007), model performance with regard to discharge at the wetland outlet (Pont de Pierre) was good to very good (NSE close to 0.75 during both calibration and validation and overall, percent bias lower than ± 10 % overall). Both high and low flows were well reproduced. Performance was not as good for the smaller upstream reaches with the model slightly underestimating total discharge and predicting flashier patterns compared to those observed. However performance was still classified as satisfactory (overall NSE > 0.5).

Model performance with regard to groundwater levels was variable (Figure 4 and Figure S2 in the Supplement). It should be assessed relative to the site's topography: the altitude range from the bottom of the mire to the top of the hills surrounding it and delineating the model boundaries is 110 m. The performance was therefore clearly good to very good for a large number of dipwells (such as D7, D13, D15, D18, D20 and D21) for which RMSEs in the order of ≤10 cm and NSEs of between 0.55 and 0.85 are achieved. In these dipwells, all located within the mire some distance from the stream and the mire margins, the

seasonal patterns were well reproduced. This was the case for instance in Dipwell D7, at the centre of the mire, where the surface saturation in winter, the drop in groundwater level in late spring or early summer and the rapid fluctuations caused by





summer precipitation events were clearly reproduced. The slight underestimation of groundwater levels in winter and spring was more a reflection of the difficulties in accurately defining ground level in peatlands (Dettmann and Bechtold, 2016) than a modelling issue (this is further discussed in Sect. 4.2).

The model performance was less good, but still satisfactory, in some dipwells (e.g. D3, D9 and D10) which are close to the
mire boundary or within the mineral soils just outside it. In D3 for instance, the much shallower groundwater table levels observed in 2012 and 2013 compared to 2011 were not very well reproduced. Conversely in D10, simulated levels were too shallow during the summers of 2012 and 2013. Dipwells D3, D9 and D10 are located where discrepancies between the actual ground elevation of the observation point and that interpolated at the corresponding DEM grid cell are the largest. This is a result of the relatively steep slopes and it is at these locations that the approximations caused by the model discretisation are
the largest. During calibration, it became evident that simulated water table depths at these dipwells were more sensitive to the parameterisation of the fissured zone than those located at the centre of the mire. While it was possible to improve the fit for dipwells on one side of the mire, this improvement was balanced by deterioration in performance for dipwells located on the other sides. This may be explained by the fact that the fissured zone was modelled as a homogeneous layer, an assumption likely to be too stringent (see Sect. 4.1).

Relatively poor performance was achieved for dipwells located close to the stream in the lower part of the mire (D8, D16, D17). Dipwells D16 and D17 are inserted into highly permeable alluvial gravel beneath or within the peat along the stream, and as a consequence groundwater table in these dipwells is largely driven by water levels in the nearby stream. The poor performance of the model in these locations was therefore a direct consequence of the absence of alluvial deposits in the model and the likely poor simulation of water levels in watercourses resulting from the kinematic routing method (Duranel, 2015).

Finally, performance was poor in Dipwell 22 as a consequence of the presence of a 30 cm deep superficial layer of oxidised peat in this part of the mire resulting from historical agricultural drainage. This layer was not represented in the model due to its small extent and as a result, the deeper and more variable observed groundwater level was not adequately simulated (Figure 4). However, both the oxidised peat and the gravel deposits occupy only a small proportion of the mire extent, and their impact on the overall model performance is therefore considered to be small.

The model predicts the observed extent of mire habitats very well. The long-term mean annual groundwater level threshold that best discriminates between mire and non-mire habitats was found to be 0.166 m below ground level. This gives a value for Cohen's kappa agreement function of 0.841, corresponding to 95.0 % of grid cells being correctly classified (Table 3, Figure 5). The good match between observed and simulated mire boundaries was not only observed within the main extent of the mire, but also along the narrow valley downstream and in the small sub-basins upstream (locations labelled (b), (d), (g)
and (h) in Figure 5). These small basins are located 10–30 m above the main mire suggesting that the model simulates groundwater levels well even in the upper part of the catchment where no groundwater level time-series were available for calibration. Small, localised discrepancies between the observed and simulated mire boundaries may be explained by different factors. Some false positives (grid cells wrongly predicted to support mire vegetation) may in part be due to errors in the mire vegetation map itself, for instance along the narrow valley in the north-east of the catchment (location (a) in Figure 5) where




difficult access and tree cover may have hindered mapping efforts. Location (c) coincides with large errors produced by the MIKE SHE overland flow component. DEM inaccuracies may explain discrepancies at locations (e) and (f) along the south-eastern margin of the mire, since this area corresponds to the boundary between the low-resolution, low-accuracy IGN BD Topo DEM on the south-eastern hillslopes and the high-resolution, high-accuracy DGPS DEM within the mire.

Overall despite these issues, the satisfactory to very good performance at a large number of monitoring locations for both discharge and groundwater table depth, and the excellent fit between observed and simulated mire boundaries, suggest that the model satisfactorily reproduces the dominant hydrological characteristics of the mire including its water balance which is discussed below.

### 3.2 Model sensitivity

Figure 6 provides the mean absolute scaled sensitivity coefficient for each model parameter that was tested with respect to groundwater level, discharge at the mire outlet (Pont de Pierre in Figure 1) and discharges of the small upstream reaches (Girolles, Marzet and Rocher in Figure 1). The parameters to which the model is most sensitive with regard to groundwater level are first and foremost those related to the hydrological characteristics of mineral formations: the horizontal saturated hydraulic conductivity, the depth and specific yield of the fissured zone, and the available water capacity of the unsaturated

mineral soils. This is the case whether dipwells are located on mineral soils or within the mire. The peat parameter to which the model is most sensitive is the vertical saturated hydraulic conductivity, which controls upwelling/downwelling fluxes between the peat layer and the underlying mineral formations.

The horizontal saturated hydraulic conductivity of the fissured zone is also the parameter to which simulated discharges of upstream reaches are by far most sensitive. Other influential parameters determine unsaturated flow in mineral soils: specific

yield, available water capacity and the bypass fraction. Discharge at the mire outlet is also strongly influenced by parameters describing fluxes in mineral formations: available water capacity and specific yield in mineral soils, and horizontal saturated hydraulic conductivity and depth of the fissured zone. Stream Manning's n is also very influential. Parameters describing the peat layer are less important.

### 3.3 Mire water balance

Figure 7 shows the simulated long-term (2001–2013) mean annual water balance of the mire (corresponding to the peat soil domain in Figure 2). Detailed water balances of both the peat and mineral soil domains are provided in Table S1 in the Supplement. The water balance computational error was 9.9 % of total annual inflows and was almost entirely due to the overland flow component. As a result of the quasi-constant saturation, very little infiltration is simulated within the mire so the overland flow component error mainly affects simulated overland outflow to the river and has little impact on other water

balance terms. The water balance error within the mineral soils domain was only 1.6 % of total inflows. This is, again, mostly caused by the overland flow component and therefore estimated inflows to the mire are not significantly impacted by model computational errors. The three main sources of water to the mire are precipitation (32.1 % of total inflows), groundwater





upwelling from the underlying mineral formations (27.1 %), and overland boundary inflow (i.e. non-channelized runoff from the mire catchment). The latter accounts for 40.2 % of total inflows, however a large proportion of this originates from edge-focused groundwater seepage on mineral soils just upstream of the mire (see Sect. 3.4). Lateral saturated boundary inflows from the mineral catchment and from the river are negligible (0.5 % and <0.0 % of total inflows respectively). Significantly,

groundwater upwelling provides 92.4 % of inputs to the mire saturated zone, which drives groundwater levels. Percolation from the unsaturated zone to the saturated zone (5.3 %) and infiltration from overland to the saturated zone (0.7 %) are only possible when the groundwater table drops below ground level for 2–3 months in summer and early autumn. At all other times, inputs from precipitation and overland boundary flow are rapidly evacuated as saturation-excess runoff and so do not contribute to the water balance of the saturated zone.

Figure 8 breaks down the mire's long-term water balance by calendar month. The largest total inflows to the mire occur in winter and early spring. They then gradually decline throughout the spring and summer reaching a minimum in September. Total outflows, dominated by overland runoff to the stream and to a lesser extent by evapotranspiration, follow the same pattern. Changes in storage are relatively small, reflecting the very shallow groundwater levels and the rapid evacuation of any water surplus through saturation-excess runoff, particularly in late autumn, winter and early spring. Due to the site's position

at the transition between a degraded oceanic climate and a mountainous climate (Joly et al., 2010), precipitation is, on average, relatively well distributed throughout the year although there is relatively large inter-annual variability. Overland boundary inflow is highly variable seasonally. It reaches its maximum in winter and early spring (accounting for 49.6 % of total inflows in January and February), before gradually declining to a minimum in September (22.9 %). It also exhibits large inter-annual variability reflecting the similar variability in precipitation. Upward saturated flow from the lower computational layer (i.e.

groundwater upwelling) is relatively constant from December to May before gradually declining to reach a minimum in September. It is characterised by much smaller seasonal and inter-annual variabilities than precipitation and overland boundary inflow and, therefore, generally provides a larger proportion of total inflows in summer and early autumn (37.2 % in September, Figure 9). As noted above, groundwater upwelling accounts for 92.4 % of total inflows to the mire saturated zone on average. Seasonally this figure peaks at 95.5 % in January and declines to 88.1 % in August. At this time receding groundwater levels

allow for the formation of an unsaturated zone across a larger proportion of the mire and for precipitation and overland boundary inflow to infiltrate and percolate, respectively, instead of producing saturation-excess runoff.

### 3.4 Spatial patterns in groundwater table depth, groundwater upwelling and groundwater seepage

Figure 10 shows the spatial distribution of simulated long-term (2001–2013) mean annual groundwater table depth and groundwater upwelling and seepage rates. As discussed above, the mire boundaries are closely associated with shallow

groundwater table depths. Groundwater upwelling and seepage rates show very similar spatial patterns and are highest along or immediately upstream of the mire boundary. This is a clear example of edge-focused discharge (Richardson et al., 2001), whereby groundwater discharge at the ground surface is enhanced where there is a break in the slope of the surface topography and, consequently, of the water table gradient. The lower topographic slope within the mire itself leads to a rapid decrease in





the vertical hydraulic gradient towards the centre of the mire, however the model suggests that groundwater seepage occurs throughout the mire. Localised discrepancies between observed mire boundaries and simulated groundwater table depth described in Sect. 3.1 are also evident in the spatial patterns of groundwater upwelling and seepage, in particular along the south-eastern boundary of the mire.

Figure 11 shows the mean spatial distribution of simulated long-term groundwater seepage for each calendar month (spatial patterns of groundwater upwelling are similar and are therefore not shown). The extent of the area where groundwater seepage occurs peaks in winter, when it extends largely to mineral soils surrounding the mire and to the small unchannelized valleys and slope breaks in the upper part of the catchment. This area progressively shrinks reaching its minimum extent in September. It is at this time of the year that the agreement between spatial patterns in simulated groundwater seepage and observed mire

boundaries is highest (kappa = 0.845). The monthly mean groundwater seepage threshold best discriminating between mire and non-mire vegetation during this calendar month is 0.005 mm day$^{-1}$, suggesting that the mire boundaries are determined by the presence or absence of groundwater seepage during the driest month of the year.

**3.5 Processes driving groundwater table depth in peat soils**

Figure 12 shows time-series of spatially averaged monthly mean groundwater table depth and selected water balance items in

the peat soils. Groundwater upwelling and overland boundary inflow are highly correlated (Spearman's rank correlation = 0.973, Figure S3 in the Supplement), the later increasing exponentially with the former, whereas both are only weakly correlated to precipitation. This further demonstrates that most of the overland boundary inflow to the mire originates from groundwater seepage through mineral soils. All terms except overland boundary inflow are significant in a full generalised additive model (Table 4). A reduced model without overland boundary inflow has a lower Akaike information criterion (AIC)

than the full model (-597.3 vs. -593.5, respectively). Given the process link between groundwater upwelling within the mire and overland boundary inflow (both related to upward movement of groundwater from the underlying granite), the later was not used for variation partitioning. Figure 13 shows the smooth terms of the reduced model. It demonstrates that, with all other variables being fixed at their sample mean, groundwater table depth increases linearly with groundwater upwelling up to about 3 mm day$^{-1}$. Thereafter it reaches a plateau approximately corresponding to ground level. Beyond this level, any additional

input is converted to groundwater seepage and overland runoff to the stream and so does not contribute to further increases in groundwater table depth. Similarly, groundwater table depth increases linearly with rainfall up to about 3 mm day$^{-1}$ before reaching a plateau. It is inversely related to evapotranspiration but only above c. 2 mm day$^{-1}$. Finally, groundwater table depth is linearly (and weakly) related to groundwater table depth in the preceding month.

Variation partitioning shows that, out of the four explanatory variables in the reduced model, groundwater upwelling explains

the largest proportion of variation in groundwater table depth overall (70.3 %, see Figure 14). However, due to collinearity, a large proportion of this variation is jointly explained by other variables, in particular groundwater table depth in the preceding month. Evapotranspiration explains 35.0 % of the variation in groundwater table depth, with 14.0 % of this being attributable to this variable independently of others.



## 4 Discussion

### 4.1 Modelling small granitic headwater catchments

Groundwater modelling in hard rock regions is often described as difficult due to the heterogeneity of the substratum (Levison et al., 2014). In the Dauges catchment, poorer model performance in some locations (e.g. D4 and D5), an apparent trade-off between performance at dipwells located on three opposite margins of the mire (D3, D9, D10, D24 and D25), and detailed analysis of ERT results and geological drilling logs (Duranel, 2015) suggest that, at the very local scale, representation of the fissured zone as an homogenous entity is an over-simplification. This is particularly the case for its hydraulic conductivity. The model was moderately sensitive to vertical saturated hydraulic conductivity of the fissured zone. Although the calibrated value was $5 \times 10^{-5}$ m s$^{-1}$, values between $1 \times 10^{-5}$ m s$^{-1}$ and $1 \times 10^{-4}$ m s$^{-1}$ produced similar results. The model was, however, highly sensitive to the horizontal hydraulic conductivity in the saturated fissured zone. This was the most important parameter controlling both stream discharge in the upper reaches and groundwater table in mineral and peat soils, and the third most important parameter controlling discharge at the Pont de Pierre mire outlet. The calibrated value was $7.5 \times 10^{-7}$ m s$^{-1}$. This is lower than the pumping test values reported by Dewandel et al. (2006) and Wyns et al. (2004), who suggested generic values of $1 \times 10^{-6}$ m s$^{-1}$ vertically and $1 \times 10^{-5}$ m s$^{-1}$ horizontally, but very similar to values estimated through model calibration by Jaunat et al. (2016) in the Pyrenees (France), Ahmed and Sreedevi (2008) in India, Koïta et al. (2013) in Ivory Coast, and Hassan et al. (2014) in Spain. All of these authors, who worked on catchments significantly larger than the Dauges catchment, found that using spatially variable hydraulic conductivities for the granite weathering formations or introducing linear vertical barriers representing assumed or observed dykes, reefs or faults was necessary to achieve satisfactory performance. At the Dauges site, the ERT survey (Duranel, 2015) highlighted large variations in the electrical resistivity of the saturated fissured zone that are most probably related to variations in granite weathering, porosity and clay content. In particular, electrical resistivity was lower and the inferred degree of weathering higher at the bottom of the etch-basin. This would agree with the current understanding of weathering and etching processes in granite landscapes (Godard et al., 2001). A homogeneous fissured layer is therefore clearly an over-simplification. Despite this, the model achieved satisfactory to very good overall performance with regard to groundwater table depths and discharge. In addition, there was very good agreement between the simulated long-term mean groundwater table depth and the observed distribution of mire habitats across the entire modelled area, including in places where no groundwater table depth observations were available for calibration. This demonstrates that, at least in some cases, saturated flow within weathered granite formations, including the fissured zone, can be reasonably well modelled at high spatial resolution using an equivalent porous medium approach and spatially homogeneous hydraulic parameters.

### 4.2 Modelling peatlands

During calibration, the bed resistance of channels represented in MIKE 11 had to be increased substantially and beyond tabulated values given in the literature (Dingman, 1994) to improve performance with regard to discharge at the Pont de Pierre





mire outlet. This reflects issues with the kinematic routing representation of discharge and stage within the mire. The low slope gradient, the shallow and ill-defined channel, the presence of shrubs and dense vegetation within the channel, and, in places, the subterranean course of the stream make the flow sluggish and create constant backwater effects. As a consequence, flooding and exchanges through highly permeable alluvial gravel deposits below the peat could not be accurately modelled, as evidenced

by the poor performance for dipwells D8, D16 and D17. However, as shown by the good performance obtained for dipwells D18 and D19, the impact of the poor simulation of stream stage does not extend beyond the area where substantial alluvial deposits occur.

Mires are frequently characterised by a hummock and hollow micro-topography caused by peat accumulation and plant growth rates that vary at a very small spatial scale (Belyea and Clymo, 2001; Malmer et al., 1994). Purple-moor grass (*Molinia*

*caerulea*) for instance can form tussocks that are up to 50 cm high. The definition of ground level is therefore a substantial methodological issue in mire hydrology (Whitfield et al., 2009), as this definition will have a profound impact on all quantities defined relative to ground level such as groundwater table depth or flood depth. Yet this definition is very rarely specified in most hydrological studies of mires. Van der Schaaf (2002) defined the surface level as "the average bottom level of the hollows within a radius of 1 m around a measuring point". In the current study the surface level adjacent to groundwater monitoring

instrumentation was taken as the mean of a minimum of three DGPS elevation measures taken within 30 cm of the dipwell and spread around it. In some dipwells this definition does not match the level at which overland flow starts to occur. This is the case for instance at dipwells D3, D7 and D12 where, except during the two or three driest months of the year, observed groundwater levels are constantly three to six centimetres above "ground level" when there is no substantial and permanent overland flow. This issue has a substantial and detrimental impact on model performance statistics but is difficult to resolve.

During model calibration, the peat specific yield and available water capacity had to be reduced substantially to 0.05 to reproduce observed groundwater table dynamics. These values are lower than those generally cited in the literature even for sapric peat. For example, based on a literature review, Letts et al. (2000) provided a mean value of 0.13 for the specific yield and 0.49 for the available water capacity of sapric peat. Specific yields similar to the value we obtained by calibration have, however, been reported elsewhere in degraded peatlands. Boelter (1968), Malloy and Price (2014) and Mustamo et al. (2016)

measured specific yields of 0.08 in decomposed peats in Northern Minnesota, circa 0.05 in a cut-over bog in Canada and 0.067 in a cultivated peatland in Northern Finland, respectively. It is well known that peat physical properties are highly variable (Boelter, 1964, 1968; Brandyk et al., 2002; Gobat, 1990; Gobat et al., 1986; Grosvernier et al., 1999), and the lower specific yield produced by model calibration at the Dauges site may result from compaction by cattle or from a higher humification rate caused by drier climatic conditions encountered in central France compared to more northern and western parts of Europe

where mires are more common and to which most of the available literature refers. The low calibrated value for available water capacity is more puzzling as it is an order of magnitude lower than those usually reported for peat soils (Letts et al., 2000). Very low values have sometimes been reported from degraded peat soils (Mustamo et al., 2016). It may also be an artefact of the two-layer unsaturated zone / evapotranspiration model, whereby the allowable range for soil moisture in the unsaturated zone has to be fully depleted before evapotranspiration can proceed from the saturated zone (DHI, 2009c) and impact



groundwater levels. In wetlands where the groundwater level is usually within the root zone, representation of transpiration as a process drawing water from both unsaturated and saturated zones simultaneously might be more realistic.

Groundwater table depths within peat soils and discharge at the Pont de Pierre mire outlet were moderately sensitive to the saturated hydraulic conductivity of the peat layer. The calibrated value was almost identical to the median value obtained from slug tests in the catotelm ($5 \times 10^{-8}$ m s$^{-1}$ vs. $4.3 \times 10^{-8}$ m s$^{-1}$), implying low to moderate permeability and compatible with the high humification rate of the catotelm suggested by the calibrated specific yield and peat stratigraphic surveys. Slightly higher values (up to $2 \times 10^{-6}$ m s$^{-1}$) gave relatively similar overall model performance but with a shift along the Pareto front towards better performance for discharge at the mire outlet and poorer performance for some dipwells. In particular, higher hydraulic conductivities led to slightly deeper groundwater table depths at the mire margins and shallower groundwater table depths at its centre. This suggests that the peat layer acts as an aquitard leading to the semi-confinement of the aquifer located in the underlying mineral formations, and to higher piezometric heads along the mire margins (see Rossi et al., 2012, for similar conclusions albeit in a different geological context). This process may have constituted a positive feedback to the mire's lateral expansion on steeper slopes and may have consequences for downstream discharge dynamics that should be further investigated.

## 4.3 Groundwater dependence of headwater wetlands in hard rock regions

This study demonstrates that groundwater upwelling from underlying mineral formations can be a quantitatively important and functionally critical element of the water balance of valley mires in granitic headwater catchments. Runoff inputs to the Dauges mire (a substantial part of which actually originates from edge-focused groundwater seepage just upstream of the mire boundaries) are quantitatively slightly larger than precipitation and groundwater upwelling. However, both runoff and precipitation do not contribute substantially to the water balance of the peat layer since they are quickly evacuated as saturation excess runoff due to the shallow water table. Water within the saturated peat overwhelmingly originates from groundwater upwelling from the underlying granite weathering formations. At the monthly time-scale, groundwater upwelling explains the largest proportion of the variation in groundwater table depth within the mire. The influence of hydrological processes occurring in mineral formations upslope of the mire on the mire's water balance is further demonstrated by the much higher sensitivity of simulated groundwater levels in the peat to parameters describing hydrodynamic properties of granite weathering formations and mineral soils compared to those describing peat soils. It is very clear that there is a high degree of hydrological connectivity between granite weathering formations within the mire catchment and the mire itself.

There is a close match between spatial patterns of simulated groundwater upwelling and seepage and the observed distribution of mire habitats. Mire habitats seem to be particularly associated with areas where the long-term mean seepage rate in September is greater than zero. Good agreement between spatial patterns in wetland vegetation and groundwater upwelling and seepage has been reported for a number of wetlands that depend on large aquifers in sedimentary contexts (Faulkner et al., 2016; Gerla, 1999; Reeve and Gracz, 2008), but much more rarely in hard rock regions without substantial sedimentary cover. As far as we are aware, the only study demonstrating this inter-relationship in a granitic context is that of Ala-aho et al. (2017).





Using an integrated physics-based hydrological model of the 3.2 km² Bruntland Burn catchment in Scotland, this previous study showed that valley bottom riparian peatlands coincided with steady groundwater upwelling and seepage which was responsible for persistent saturated conditions and overland flow generation even during dry periods. However, only the glacial drift sediments were considered to be hydrologically active, while the underlying granite bedrock was assumed to be

impermeable. Drexler et al. (2013) showed that the areal extent of small mountainous fens in the Sierra Nevada (USA), located on a range of geological substrates including hard rocks, had decreased by between 10 % and 16 % over the last 50–80 years. They tentatively related these changes to a simultaneous increase in temperature and decrease in snowpack volume and longevity that together reduced groundwater recharge and, in turn, seepage to the fens. It was suggested that small mountainous fens might be used as whole-ecosystem gauges of groundwater recharge through time. The Dauges modelling study supports

this assertion and changes in the factors impacting groundwater recharge within the mire catchment will likely impact spatial and temporal patterns of groundwater upwelling and seepage and, therefore, the distribution of mire vegetation. How quickly changes in groundwater upwelling rates would translate into modifications to mire vegetation is unknown. In contrast to mires dependant on large aquifers in sedimentary contexts, the hydrogeological catchment of headwater valley mires in hard rock regions is likely to be relatively small in extent due to the relative shallowness of weathering formations compared to the

surface relief. As a consequence, these mires are likely to be more sensitive to small-scales patterns and changes in groundwater recharge. As such, they may constitute useful whole catchment indicators to disentangle the respective impacts of local factors (changes in land use for instance) and regional factors (climate change in particular). This study only related spatial patterns in groundwater upwelling and seepage to the distribution of mire habitats as a whole. Future work could investigate in more detail whether groundwater upwelling and seepage rates are significant factors in explaining the variability in the species

composition of headwater valley mires in hard rock regions (Larocque et al., 2016; Munger et al., 2014).

The findings from this study have important management and legal implications. Indeed, the potential for anthropogenic pressures on groundwater bodies to indirectly impact groundwater-dependent terrestrial ecosystems (GWDTE) is now recognised in water resource legislation in many countries. In the European Union for instance, the Water Framework Directive (2000/60/EC) requires member states to ensure that water bodies, including groundwater bodies, achieve 'good ecological

status' which comprises a combination of good chemical and quantitative condition. Annex V of the Directive states that a groundwater body cannot achieve good status if the water table level is subject to anthropogenic alteration that would result in significant damage to terrestrial ecosystems that depend directly on the groundwater body. In practice the status of potential GWDTEs, in particular in hard rock upland regions, is rarely or only superficially considered due to lack of understanding of the 1) impact of anthropogenic pressures on upland hard rock aquifers; 2) hydrological connectivity between hard rock aquifers

and terrestrial ecosystems; and 3) hydrological controls on ecological conditions (Vernoux et al., 2010; Whiteman et al., 2010). Our study demonstrates that the Dauges mire depends directly on groundwater upwelling from the underlying mineral formations. Even though further work is required to upscale our conclusions, there is no reason to believe that the Dauges mire and its catchment are a unique case among valley mires in granitic headwaters. On the contrary, the Dauges site is representative of valley mires across the granitic Massif Central, and shares many similarities with other such systems found





across the world, in particular throughout the Variscan Belt (Etlicher, 2005; Godard et al., 2001; Valadas, 1984, 1998). The results of this study show that the ecohydrological conditions of valley mires must be considered when assessing the status of hard rock groundwater bodies under the Water Framework Directive.

In common with the Dauges mire, many headwater valley mires in hard rock regions are designated for nature conservation under national or international legislation (Muller, 2018). Anthropogenic pressures on groundwater recharge within the catchments of these mires must be considered when defining conservation measures for these habitats or undertaking environmental impact assessments of policies and planning decisions. Article 6 of the EU 92/43/EEC Habitats Directive states that "any plan or project […] likely to have a significant effect thereon, either individually or in combination with other plans or projects, shall be subject to appropriate assessment of its implications for the site in view of the site's conservation objectives". The present study demonstrates that any activities likely to result in substantial changes in groundwater flow within granite weathering formations upstream of the mire, even beyond SAC boundaries, do fall within the remit of this article. Examples of such activities include changes in land use such as large-scale afforestation and groundwater abstraction. Afforestation has, for example, been widespread within the Massif Central (Derrière et al., 2013; Dodane, 2009). The resulting changes in interception and evapotranspiration, with consequent impacts on surface and subsurface hydrological flow paths (Andréassian, 2004; Farley et al., 2005; Katzensteiner et al., 2011), may have had impacts on the region's mire ecosystems that should be further investigated.

Assuming that the relative importance of precipitation, runoff from mineral soil and groundwater seepage in the mire's overland component water balance reflects their relative contributions to overland inputs to water courses, 23 % of stream discharge at Pont de Pierre is water originating from the underlying mineral formations that has seeped through the peat layer. Excluding evapotranspiration, 46 % of water that flowed out of the saturated zone through seepage to overland flow, direct seepage to watercourses and groundwater flow out of the area upstream of Pont de Pierre did so through the peat layer. The vast majority (96%) of the stream discharge at Pont de Pierre flowed at some point through the mire, either as overland flow or saturated flow. These figures highlight the importance of the mire as an interface between granitic weathering formations and watercourses, and hint at its potential role in mitigating the transfer of potential harmful compounds such as arsenic or uranium derived from the local geology (Mauroux et al., 2009) to water courses. Due to their organic soils and reductive conditions, pristine mires have been shown to constitute major sinks for almost all metallic and metalloid contaminants, and to efficiently reduce loads in downstream watercourses (Brown et al., 2000; González A et al., 2006; Lidman et al., 2012; Sobolewski, 1999). However most of these studies have focussed on load reduction in watercourses as they flow through riparian mires, and further research is needed on load reduction in groundwater as it seeps through groundwater-fed mires.

## 5 Conclusions

Using an integrated MIKE SHE / MIKE 11 hydrological / hydraulic model, this study has shown that, contrary to the 'impermeable bedrock dogma' that has long been the dominant view in hard rock hydrogeology and hillslope hydrology,



groundwater upwelling from granite weathering formations is a quantitatively important and functionally critical element of the water balance of the Dauges mire, a headwater valley mire in the granitic uplands of the French Massif Central. Model performance in terms of simulated water table levels and stream discharge was at least satisfactory and in many locations good to very good. It included the replication of the seasonal rise and fall in groundwater levels. Based on model simulations,

groundwater upwelling provided 27.1 % of total long-term inflows to the mire, rising to 37.2 % in September when total inflows are small. Furthermore, groundwater upwelling provided 92.4 % of total long-term inflows to the mire saturated zone. Variation partitioning showed that groundwater upwelling explained 70.3 % of the variation in monthly mean groundwater levels within the mire, although a large proportion of this is shared variation jointly explained by other variables, in particular groundwater table depth in the preceding month. The distribution of mire habitats within the catchment appears to be strongly

controlled by groundwater upwelling. It was shown to closely coincide with areas where the simulated long-term mean groundwater seepage rate in September, the driest month associated with the lowest water table levels, is greater than zero. Model results demonstrate that the Dauges mire is a groundwater-dependent system closely connected to the granitic aquifer. This has important implications for the management and conservation of this and other similar ecosystems. However, whilst the Dauges mire is representative of acidic valley mires commonly found in many Hercynian uplands, further work is required

to upscale the conclusions of this study, and to investigate how groundwater upwelling rates influence the ecology and biogeochemistry of such systems.

## 6 Code availability

MIKE SHE / MIKE 11 is commercial modelling software produced by DHI. Demo versions are available from www.dhigroup.com.

## 7 Data availability

With the exception of secondary data derived under licence from third-party datasets, data used for this study are available upon reasonable request from the first author (arnaud.duranel@univ-st-etienne.fr). The meteorological data can be obtained from Météo-France, http://donneespubliques.meteofrance.fr.

## 8 Supplement link

## 9 Author contribution

AD designed and managed the study, installed the hydrometeorological network, collected all the data with the exception of manual checks, developed the geological conceptual model and MIKE SHE / MIKE 11 models, interpreted the results and





wrote the first draft of the manuscript. JRT, HB and HC were AD's PhD supervisors and provided advice on the research design and on data analysis. JRT provided substantial expert advice on MIKE SHE / MIKE 11 and edited the initial manuscript. HC was principal investigator on the EFRD and AELB grants that funded the hydrometeorological equipment and most field expenses. PD manages the Dauges National Nature Reserve. He contributed to the maintenance of the hydrometeorological network, completed regular manual checks, and provided substantial technical and logistical support during field work. SG provided the ERT equipment, processed the ERT data and contributed to their interpretation. RW made a major contribution to the interpretation of the geological and geomorphological data and to the development of the geological model.

## 10 Competing interests

The authors declare that they have no conflict of interest.

## 11 Acknowledgements

This work was supported by a UK Natural Environmental Research Council (NERC) doctoral studentship to AD [grant number NE/H525203/1] and a grant to UMR 5600 CNRS EVS from the European Fund for Regional Development (EFRD) and the Agence de l'Eau Loire-Bretagne (AELB), obtained as part of the Plan Loire Grandeur Nature managed by the Etablissement Public Loire (http://www.plan-loire.fr), with additional contributions from the University of Saint-Etienne. The Conservatoire des Espaces Naturels de Nouvelle-Aquitaine and the Réserve Naturelle Nationale de la Tourbière des Dauges dedicated staff time to the collection of data used to develop the hydrological model. Météo-France provided part of the meteorological data. The Institut National de l'Information Géographique et Forestière, Areva (now Orano), the Conservatoire des Espaces Naturels de Nouvelle-Aquitaine and the Réserve Naturelle Nationale de la Tourbière des Dauges provided geological, topographic and cartographic data.

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



**Figures**



**Figure 1. The Dauges mire, its catchment and the hydrological monitoring network.**





**Figure 2. Water balance domains used within the MIKE SHE / MIKE 11 model of the Dauges catchment.**





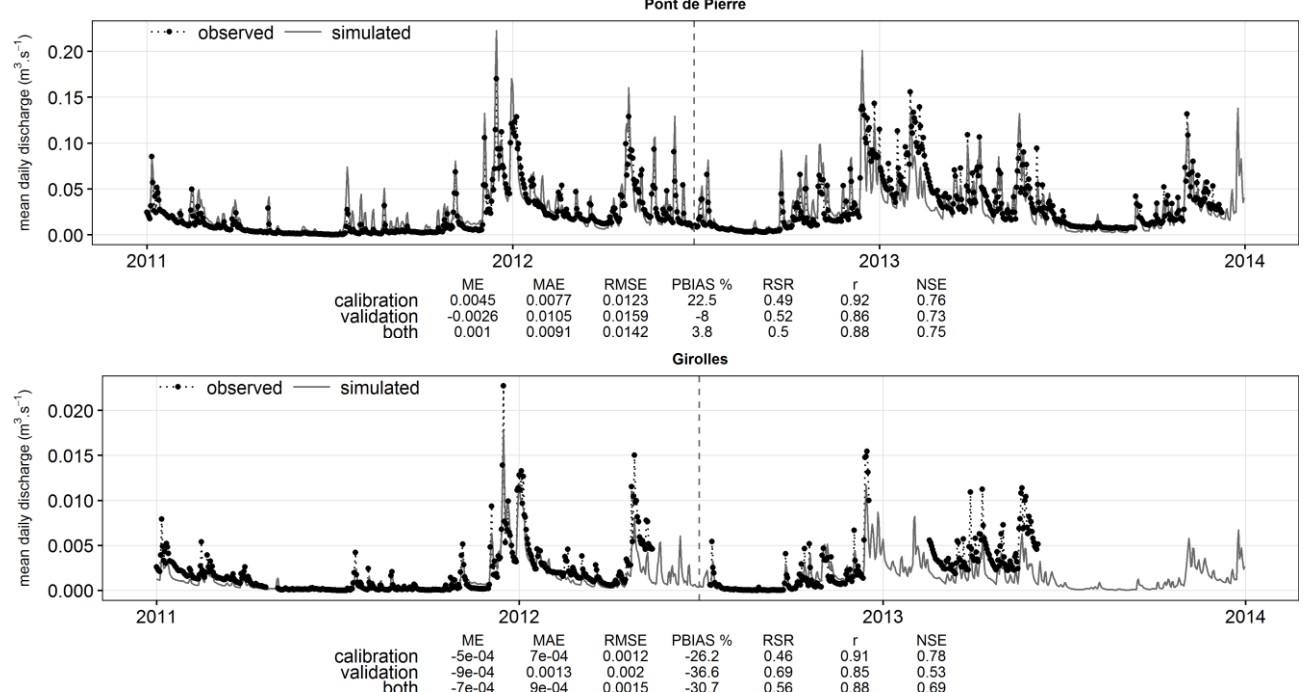

**Figure 3. Observed and simulated stream discharge and model performance statistics for two selected locations within the Dauges catchment (01/01/2011–31/12/2013). Note different y-axis ranges.**





**Figure 4. Observed and simulated groundwater table depth and model performance statistics for selected dipwells within the Dauges catchment (01/01/2011–31/12/2013). Note different y-axis ranges.**









5    **Figure 4 (continued). Observed and simulated groundwater table depth and model performance statistics for selected dipwells within the Dauges catchment (01/01/2011–31/12/2013). Note different y-axis ranges.**

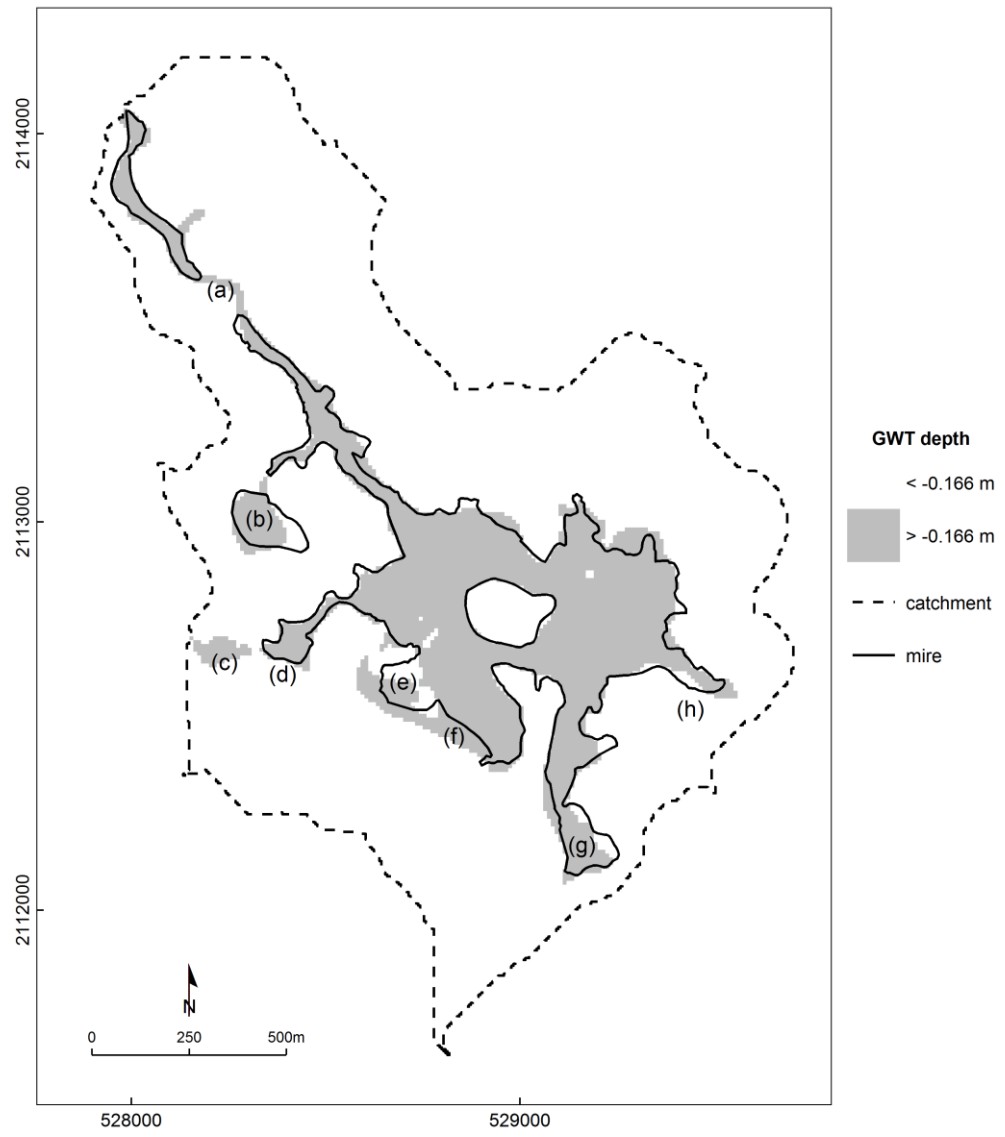

**Figure 5. Observed mire boundary based on botanical (Durepaire and Guerbaa, 2008) and pedological (Duranel, 2015) criteria and the predicted mire distribution based on model results (simulated 2001–2013 mean groundwater table depth higher than 0.166 m below ground level).**



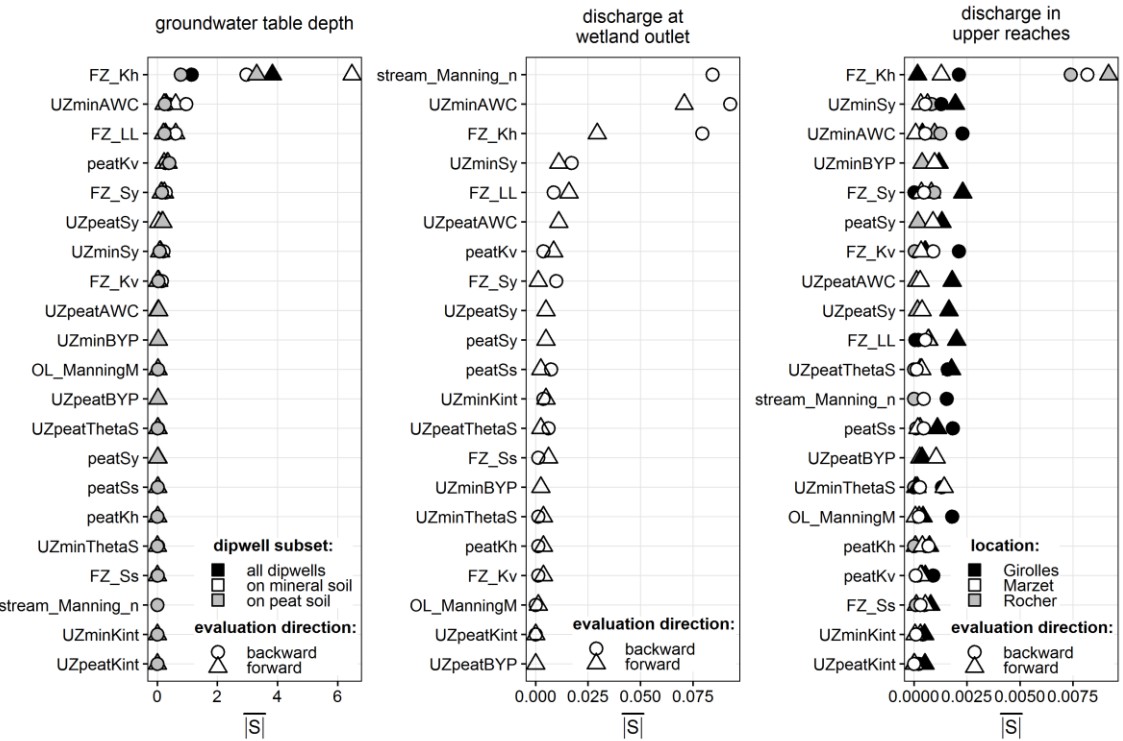

**Figure 6. Summary of model sensitivity analysis results (see Table 2 for parameter codes). Parameters are arranged in order of increasing mean sensitivity from bottom to top.**



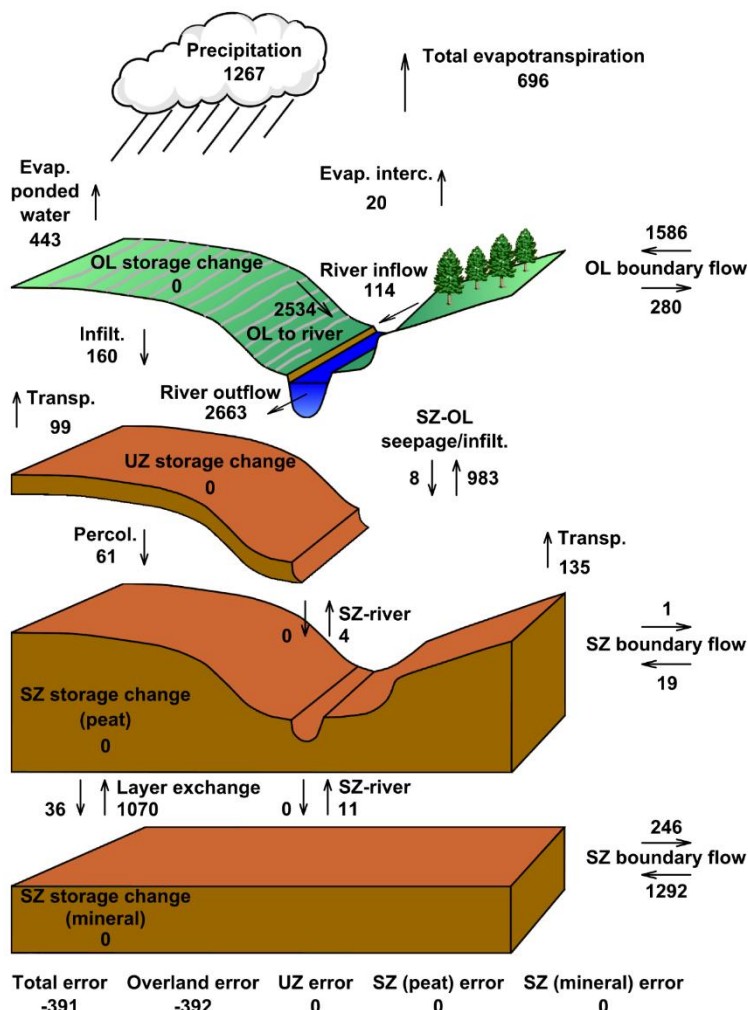

**Figure 7. Simulated mean annual water balance of the Dauges mire (mm yr⁻¹) for the period 2001–2013. Evap.: evapotranspiration; interc.: interception; transp.: transpiration; infilt: infiltration; percol.: percolation; OL: overland; UZ: unsaturated zone; SZ: saturated zone.**







**Figure 8. Simulated mean monthly water balance of the Dauges mire (2001–2013). Figures follow the MIKE SHE water balance convention: inputs are negative, outputs positive, change in storage is positive when storage increases, and the water balance error is the sum of all inputs and outputs. Whiskers show the standard deviation.**



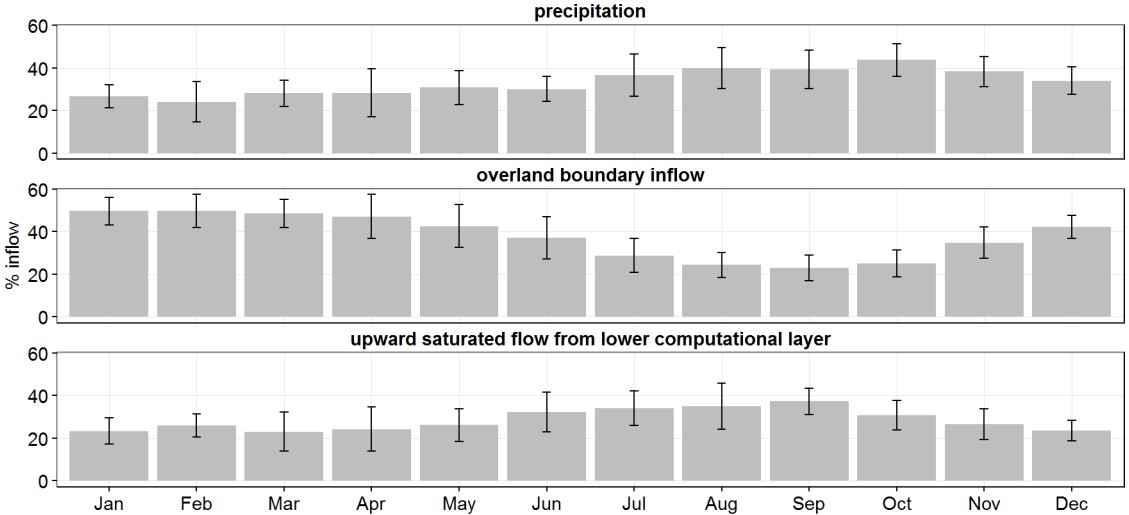

**Figure 9. Mean monthly proportion of simulated inflow to the Dauges mire from different sources (2001–2013). Whiskers show the standard deviation. Lateral saturated boundary flow from the mineral catchment and from the river account for less than 0.6 % of total inflows at all times and are therefore not shown.**





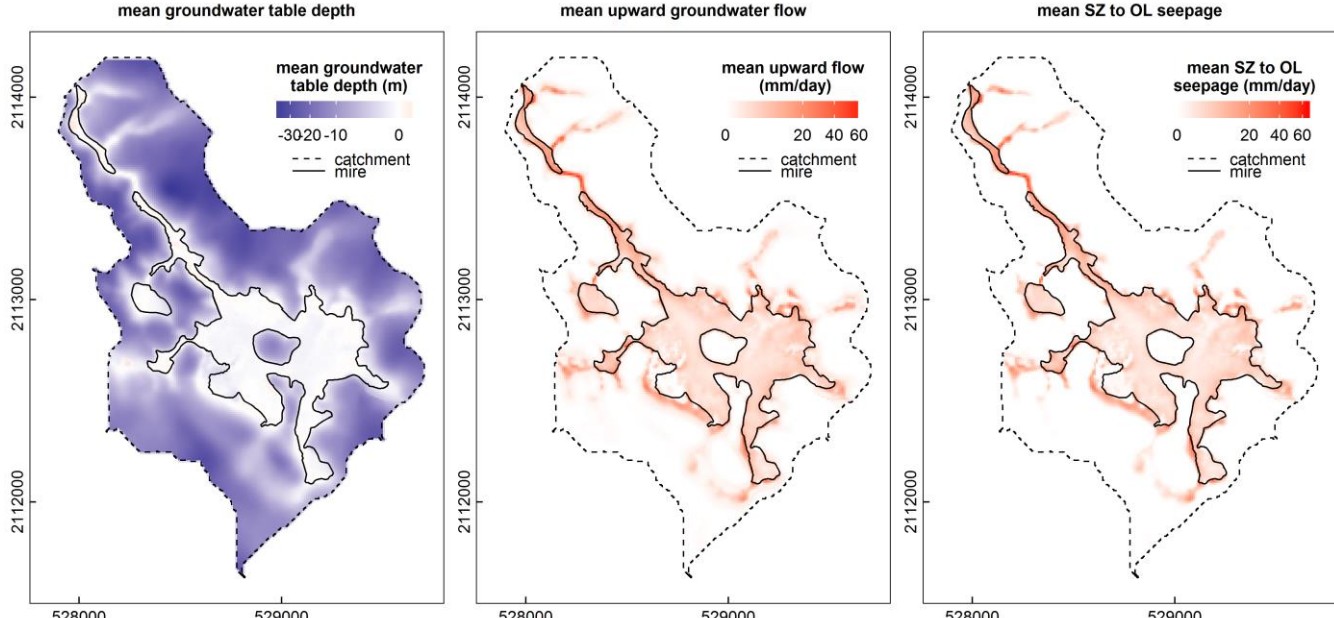

**Figure 10. Simulated mean annual groundwater table depth, upward groundwater flow and seepage rate (2001–2013).**



**Figure 11. Simulated mean monthly groundwater seepage rate (2001–2013).**

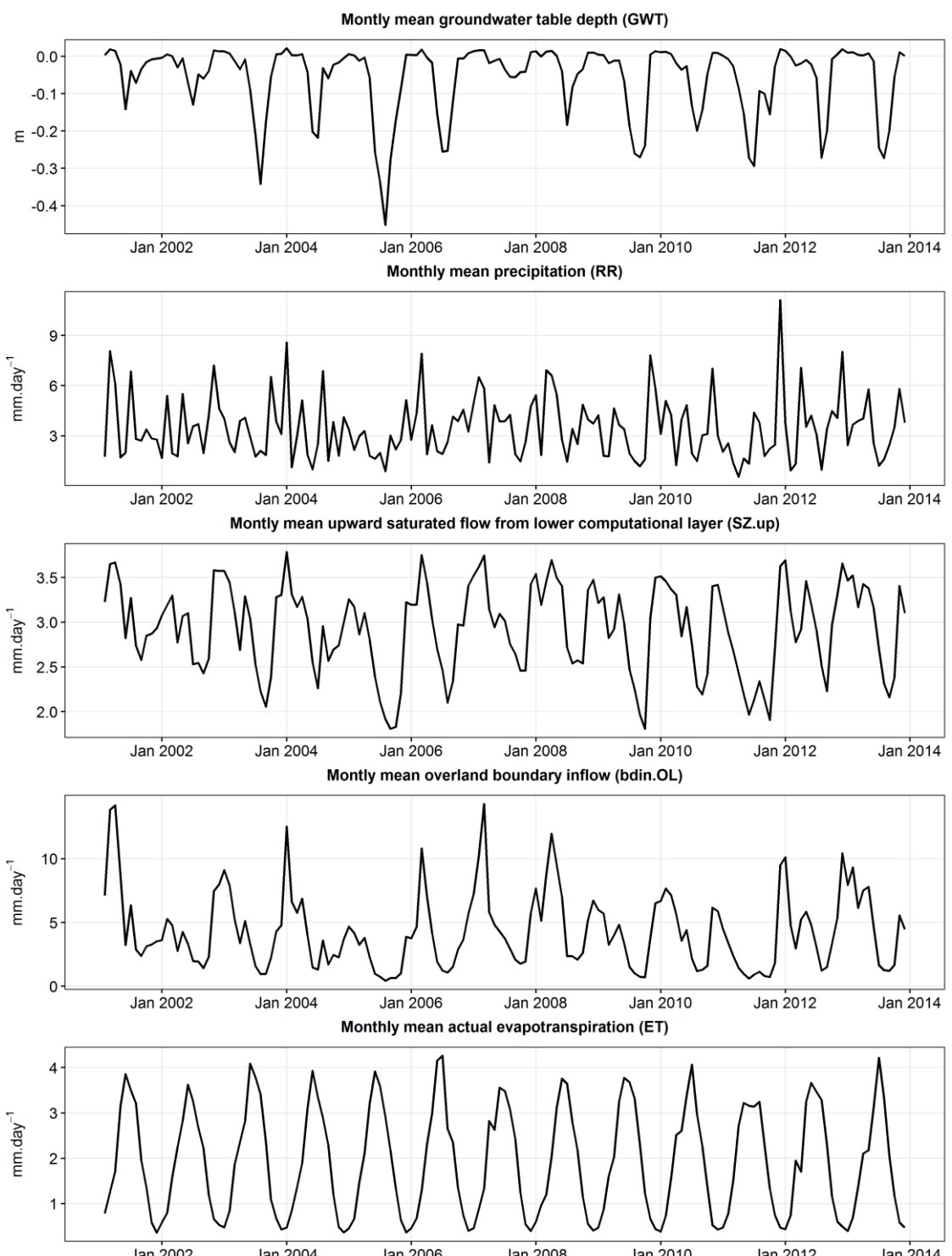

**Figure 12. Monthly means of simulated groundwater table depth and selected water balance items in the peat soil domain (2001–2013).**

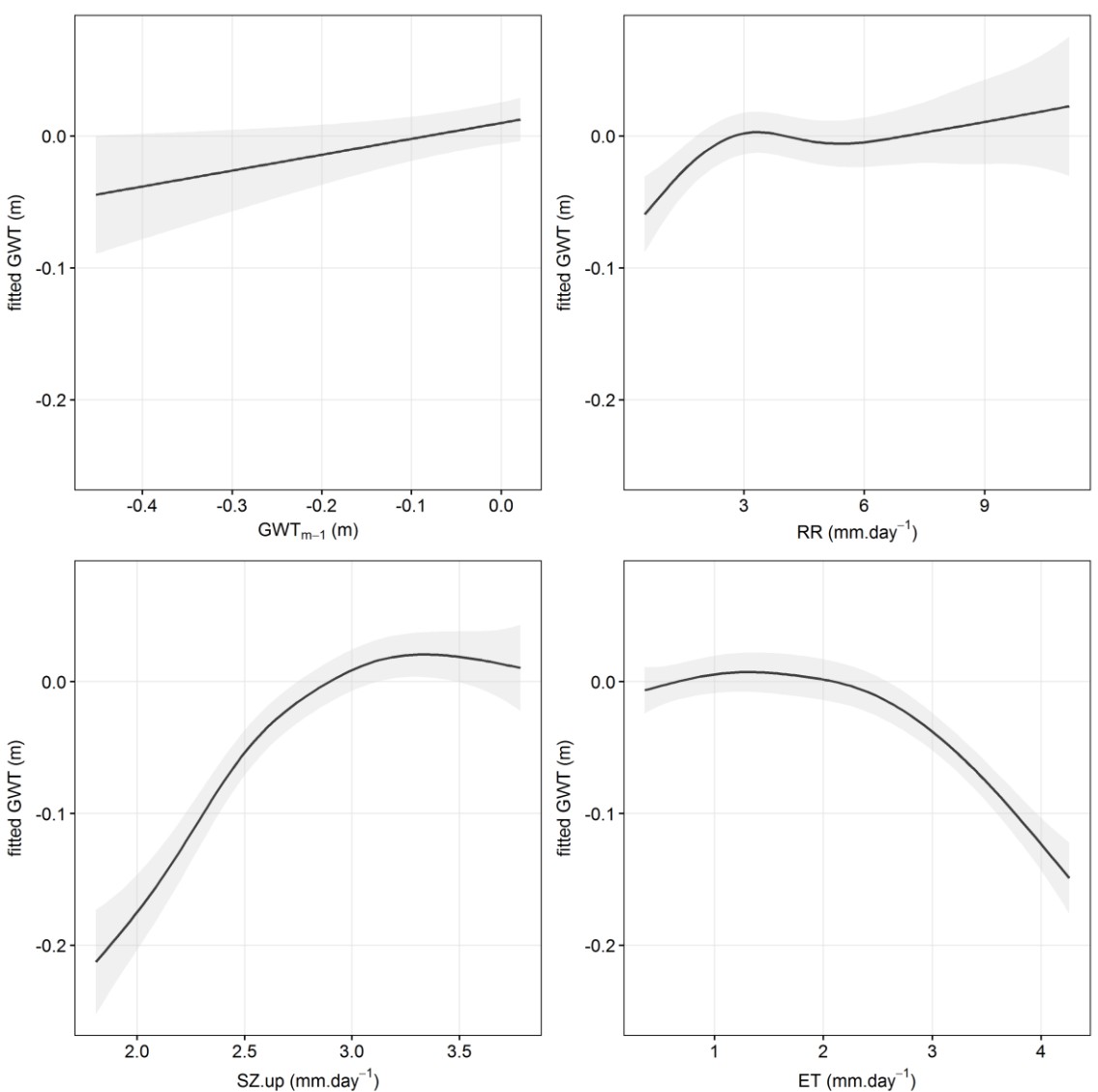

**Figure 13. Smooth terms of the final generalised additive model of simulated monthly mean groundwater table depth on the response scale. The curve shows the fitted value of the response, conditional upon the other explanatory variables being held at their sample mean. The shaded area is the approximate 95 % confidence interval.**





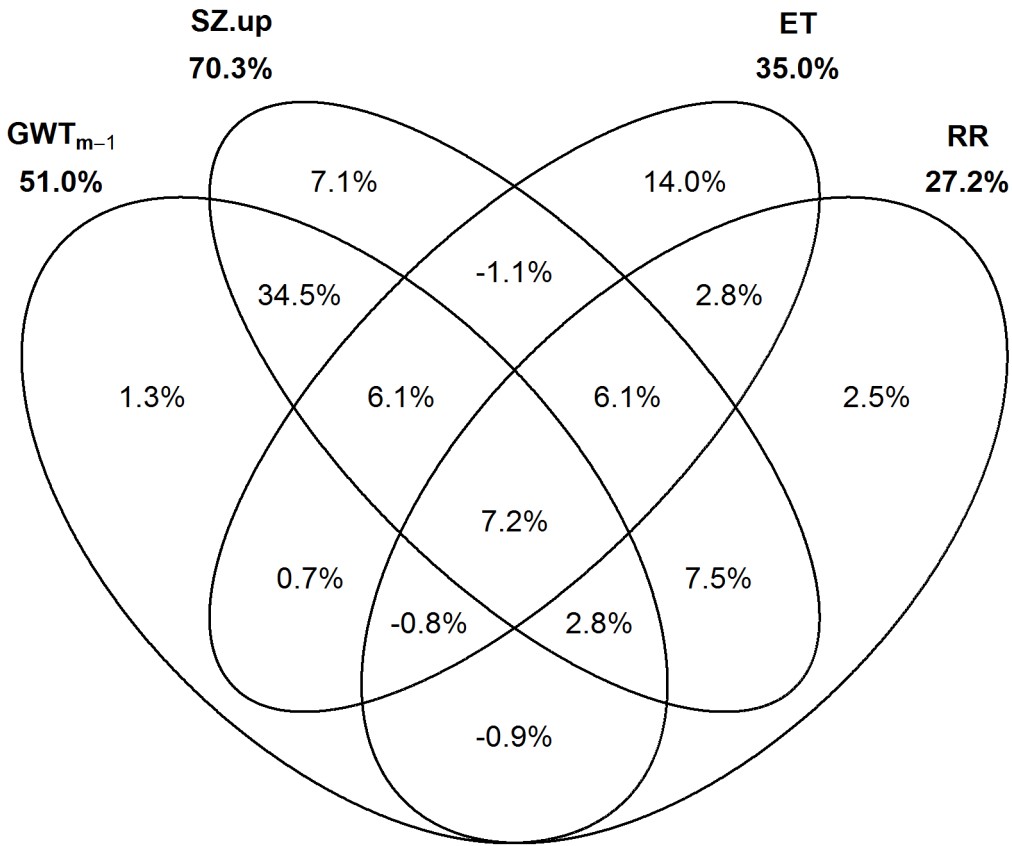

**Figure 14. Venn diagram of the variation partitioning results. The diagram shows the proportion of deviance in groundwater table depth (GWT) explained independently or jointly by groundwater table depth in the preceding month (GWT$_{m-1}$), upward saturated flow from the lower computational layer (SZ.up), actual evapotranspiration (ET) and/or precipitation (RR), based on simulated spatially-averaged monthly means in the peat soil domain for the period 2001–2013.**



**Table 1. Parameterisation of the evapotranspiration model.**

| | maximum leaf area index | minimum leaf area index | summer root depth (m) | winter root depth (m) | summer canopy height (m) | winter canopy height (m) | mean leaf resistance (s.m⁻¹) | winter crop coefficient (Kcb ini) | summer crop coefficient (Kcb full) | winter bulk interception fraction | summer bulk interception fraction | winter interception capacity (γ) | summer interception capacity (γ) |
|---|---|---|---|---|---|---|---|---|---|---|---|---|---|
| coniferous woodlands (Douglas fir) | 8.5 | 8.5 | 2 | 2 | 30 | 30 | 400 | 0.60 | 0.78 | 0.37 | 0.37 | 4.51 | 4.51 |
| deciduous woodlands (beech, oak and chestnut tree) | 5.8 | 1 | 2 | 2 | 24 | 24 | 400 | 0.38 | 0.77 | 0.16 | 0.28 | 1.41 | 2.81 |
| mixed woodlands | 7.2 | 4.8 | 2 | 2 | 27 | 27 | 400 | 0.59 | 0.78 | 0.27 | 0.33 | 2.85 | 3.63 |
| wet woodlands | 3 | 1 | 1.5 | 1.5 | 11 | 11 | 200 (400 in winter) | 0.40 | 0.88 | 0.16 | 0.22 | 1.41 | 1.99 |
| heath and shrubs | 2.5 | 2.5 | 0.6 | 0.6 | 0.3 | 0.3 | 300 | 0.60 | 0.69 | 0.16 | 0.16 | 1.36 | 1.36 |
| pastures and meadows | - | - | 0.9 | 0.9 | - | - | - | 0.6 | 0.9 | - | - | - | - |
| mire | - | - | 0.6 | 0.15 | - | - | - | 1.05 | 1.05 | - | - | - | - |
| impervious | - | - | 0 | 0 | - | - | - | 1.05 | 1.05 | - | - | - | - |





**Table 2. Calibration parameters of the MIKE SHE / MIKE 11 model of the Dauges catchment including their final values and the range used in sensitivity analyses.**

| Parameter | Code | Calibrated value | Range used in sensitivity analyses |
|---|---|---|---|
| **Channel flow (MIKE 11)** | | | |
| bed resistance (Manning's n) | stream_Manning_n | 0.5 | 0.025-0.5 |
| **Overland flow** | | | |
| resistance (Manning's M) | OL_ManningM | 10 | 5-50 |
| storage detention (mm) | - | 1 on mineral ground, 3 on peat soils | not tested |
| **Unsaturated zone** | | | |
| peat water content at saturation | UZpeatThetaS | 0.8 | 0.01-0.95 |
| peat specific yield (UZ) | UZpeatSy | 0.05 | 0.01-0.99 |
| peat available water capacity | UZpeatAWC | 0.05 | 0.01-0.99 |
| peat saturated hydraulic conductivity (UZ) | UZpeatKint | $2e-6$ m s$^{-1}$ | $1e-10 - 5e-4$ m s$^{-1}$ |
| peat bypass max fraction | UZpeatBYP | 0 | 0-1 |
| mineral soil water content at saturation | UZminThetaS | 0.8 | 0.01-0.95 |
| mineral soil specific yield (UZ) | UZminSy | 0.1 | 0.01-0.99 |
| mineral soil available water capacity | UZminAWC | 0.69 | 0.01-0.99 |
| mineral soil saturated hydraulic conductivity (UZ) | UZminKint | $1e-4$ m s$^{-1}$ | $1e-8 - 5e-4$ m s$^{-1}$ |
| mineral soil bypass max fraction | UZminBYP | 0 | 0-1 |
| **Saturated zone** | | | |
| fissured zone lower level (below ground) | FZ_LL | Fixed: -55 m | -100 - -5 |
| fissured zone horizontal hydraulic conductivity | FZ_Kh | $7.5e-7$ m s$^{-1}$ | $1e-7 - 5e-4$ m s$^{-1}$ |
| fissured zone vertical hydraulic conductivity | FZ_Kv | $5e-5$ m s$^{-1}$ | $1e-7 - 5e-4$ m s$^{-1}$ |
| fissured zone specific yield | FZ_Sy | 0.015 | 0.001-0.1 |
| fissured zone specific storage | FZ_Ss | $1e-5$ m$^{-1}$ | $1e-7 - 1e-4$ m$^{-1}$ |
| peat horizontal hydraulic conductivity | peatKh | $5e-8$ m s$^{-1}$ | $1e-8 - 5e-4$ m s$^{-1}$ |
| peat vertical hydraulic conductivity | peatKv | $5e-8$ m s$^{-1}$ | $1e-8 - 5e-4$ m s$^{-1}$ |
| peat specific yield | peatSy | = UZpeatSy | 0.01-0.99 |
| peat specific storage | peatSs | 0.01 m$^{-1}$ | 0.001-0.05 m$^{-1}$ |





**Table 3. Confusion matrix between the observed mire distribution based on botanical (Durepaire and Guerbaa, 2008) and pedological (Duranel, 2015) criteria and the predicted mire distribution based on model results (simulated mean annual groundwater level > 0.166 m below ground level for the period 2001–2013), in % of the total number of grid cells.**

|  |  | observed | |
| --- | --- | --- | --- |
|  |  | non-mire | mire |
| predicted | non-mire | 77.7 % | 1.4 % |
|  | mire | 3.6 % | 17.2 % |





**Table 4. Approximate significance of smooth terms in the full generalised additive model.**

| smooth term | edf | F | p-value |
| --- | --- | --- | --- |
| $s(GWT_{m-1})$ | 1.000 | 6.296 | 0.0132 |
| $s(RR)$ | 4.274 | 6.048 | $9.62 \ 10^{-05}$ |
| $s(SZ.up)$ | 4.303 | 34.87 | $<2 \ 10^{-16}$ |
| $s(bdin.OL)$ | 1.000 | 0.250 | 0.6176 |
| $s(ET)$ | 3.849 | 40.631 | $<2 \ 10^{-16}$ |

$GWT_{m-1}$: monthly mean groundwater table depth in the preceding month; RR: monthly mean precipitation; SZ.up: monthly mean upward saturated flow from the lower computational layer; bdin.OL: monthly mean overland boundary inflow ET: monthly mean actual evapotranspiration.