# Peer review of "Modelling the hydrological interactions between a fissured granite aquifer and a valley mire in the Massif Central, France"

_Hydrology and Earth System Sciences, 2020_

## Referee Comment (RC1) · Anonymous Referee #1 · 2 Jun 2020

Summary

This manuscript presents an integrated hydrological modeling study for a small wetland in France. The wetland is located in a weathered granite bedrock basin. Groundwater flow in the weathered zone is described using an equivalent porous medium approach. There is limited novelty in the applied modeling techniques. It seems that the authors see the main generic contribution of the study as a demonstration of the limits of the "impermeable bedrock paradigm". However, if that is the main purpose of the study, I feel that wetland processes should have been simulated with a multiple model approach, using different and competing conceptualizations of hydrologic processes. I do

not recommend publication of this article in HESS in its current form.

Review comments

1.) The modeling work presented in this study is technically sound but does not go beyond the state of the art. The focus of the article is clearly the simulation and improved understanding of this local wetland, which may be of limited interest for a broad international readership. Authors provide more generic context in terms of the "impermeable bedrock paradigm". However, in order to systematically analyze the importance of groundwater processes and investigate the impact of representing groundwater processes in different ways, I feel that authors should have chosen a multiple model approach and should have compared different (and competing) conceptual models of this wetland.

2.) Calibration and validation periods are fairly short. Also, a manual trial-and-error calibration is used. Are the authors confident that the calibration result is robust, i.e. performance will be comparable for new periods and scenarios? For instance, p 10 line 10 ff points out that the assumption of uniform hydraulic conductivity throughout the fissured zone may be overly simplistic. . .

3.) Section 3.2. It is important to stress that what is described here is a local sensitivity analysis (around the starting parameter values?). Sensitivity may be quite variable across the parameter space, for a complex integrated hydrological model.

Details

- p. 4, line 22ff: land elevation was measured manually on a 5x5 m grid? This sounds quite time consuming. . . maybe UAS lidar or photogrammetry would have been an efficient alternative? Also other elevation data sources seem to be quite coarse and low accuracy. Is there no highres, high quality DEM available for this region?

- p. 5, line 13ff: ERT can probably not directly map fissures, due to insufficient spatial resolution? I guess the fissuring was interpreted from lower bulk resistivity in the ERT

sections?

- p. 7: It may make sense to briefly introduce the terms acrotelm and catotelm.

- p. 11, line 27: A water balance error of 9.9% is quite high – maybe the convergence criteria should have been set tighter?

---

## Referee Comment (RC2) · Anonymous Referee #2 · 10 Jun 2020

General comment:

The manuscript studies an interesting case site in French hill regions where a peatland is situated on a valley. The question is interesting: why is the peatland at this specific place? Can we understand the hydrogeological surroundings to better explain where the water to the peatland is originating from to better protect these kind of important ecosystems and possible sources of water to them.

The approach the authors have used is an integrated MIKE model combined with rather comprehensive field measurement campaign for model calibration and validation. The specific interest of the authors is to understand the role of the weathered granite formation in the catchment as the source of the water to peatland.

Considering that the main hypothesis that the authors are studying is the role of the granite fissures, the authors do not give enough geological details how they have ended up building their model layers. The saturated zone of the model comprises of two computational layers. At peatlands the layer was 1) the peat as top layer and 2) fissured granite below; At mineral soils 3) 2 meter thick layer on top and 4) fissured granite below. If I followed the information given in manuscript correctly all of the layer structures 2-4 had the same parameters (e.g. hydraulic conductivity). Given that there were some soil on top of the granite at least locally, all of the geological information outside the peat layer are within the parameters of layers 2-4. It would be crucial to explain more in details and show the geological formations in detail that this model structure is acceptable: are the soil formations above spatially neglectable. Even if no detailed information on the soil layers are available the approximations should be shown, rather than only explaining and referring to previous source as this is the key question in the article. Also there are information in the text on the previous drillings by nuclear company, these data in addition to the available details on soil should be represented in maps/conceptual cross-sections.

As the layers 2-4 are representing generally the whole surrounding geology of the peatland the authors have to give better explanation to the geology behind the model. Currently the text leaves a possibility for model equifinality: the same model end results can come either if i) granite is fissured and explains the flow to peatland or ii) within the same layer parameters for the whole catchment there is combination of soil layers with regionally higher k and granite fissures with lower k. This general problem in the article has to be resolved and would need some major revision e.g. explaining the geology and discussing the possible uncertainties considering the model structure.

Specific comments:

Even though this bit contradicts the general main comment above (as there is a need

for more information on geology) there is a quite high amount of figures in the current manuscript. Are these all necessary and giving reader information that cannot be presented in the text or can some of the figures be combined? For example: - Is figure 9 needed in addition to figure 8?

- In figure 10 the two latter figures look identical, are both needed? Same with Figure 11, most of the months are identical and reader doesn't get a lot of information when looking them. For example: Could Figure 11 be combined to one map which shows maximum and minimum months considering seepage rate with different colors? Then moving last map away from figure 10 and combining condensed information from 10 and 11.

- Figure 12 is not opening up easily for the reader. Maybe marking a specific point in time of interest would help (e.g. with dashed vertical line)

Detailed comments:

page 4, line 29-30. Where are these boreholes on the map? Page 16, line 20: is the word evacuated the correct term? Figure 5. text. Explain what are the a-h standing for? Figure 13 text: explain abbreviations in x- and y-axes

―――――――――――――――

---

## Author Comment (AC1) · 7 Aug 2020

We thank Anonymous Referee #1 for their comments and for suggestions on the original paper. We believe that the paper will be improved by responding to the issues that they raise and by incorporating their suggested revisions.

Summary and Comment 1:

The referee suggests that we "see the main generic contribution of the study as a demonstration of the limits of the "impermeable bedrock paradigm"". Although our study does indeed contradict this paradigm, we do not see this as its main contribution, nor

its main objective. The "impermeable bedrock paradigm" is only referenced in the Introduction and Conclusion of the paper, to frame our research within the wider context of hydrological research within hard rock regions. It is not central to the aims of the study and the methods employed which, as discussed below, focus on mire hydrology. Nevertheless, we do acknowledge that issues related to hard-rock hydrogeology figure prominently in our introduction. We felt this was necessary to set the scene for our research questions and highlight the novelty of our findings; but we appreciate this may give the reader the impression that our study focuses on this issue. Therefore, in revising the manuscript, we propose to recast the Introduction to focus more explicitly on the central issues related to the hydrology of the type of wetland considered in our study. Given this, reference to the impermeable bedrock paradigm will be maintained but the central focus – i.e. modelling the hydrology of wetlands in such areas – will be much more explicit.

The referee suggests that wetland processes could have been simulated with a multiple model approach, using different and competing conceptualizations of hydrological processes. While we fully agree that such an approach would have been appropriate if our objectives had been to test the "impermeable bedrock paradigm" or to instead investigate the implications and associated uncertainty of using different process descriptions (e.g. Thompson et al., 2009, 2004), we argue the referee proposes a completely different study to that which the paper reports. Instead, our objectives, which are described at the end of the Introduction, clearly focus on the hydrology of the acidic valley mire. Using a case study, we aimed to (1) test the ability of an equivalent porous medium approach, with limited data on the hydrodynamic properties of the granite weathering formations, to reproduce high-resolution spatial and temporal patterns in groundwater seepage and groundwater table depth within the mire, (2) quantify the mire water balance including its dependence on groundwater inflows from granite weathering formations, and (3) investigate the hydrological processes driving groundwater table depth in the mire. To do this we used the best available conceptual understanding of the research catchment and its mire, supported by a large set of field data (further

described in Duranel (2015) and, in response to a suggestion from Referee #2, to be expanded upon in the revised paper). We demonstrate that groundwater upwelling can be a quantitatively important and functionally critical element of the water balance of valley mires in granitic headwater catchments.

Whilst our research is a case study and uses established methods, to our knowledge this is the first integrated hydrological modelling study of a mire that accounts for groundwater flow in weathered hard rocks, and shows a close match between spatial patterns of simulated groundwater upwelling and seepage and the observed distribution of mire habitats. Ala-aho et al. (2017) used similar methods in a similar environment, but assumed that the underlying granite bedrock was impermeable. As such, our results are novel and are of interest for a broad international readership since they have important implications for the hydrological understanding, management and conservation of such wetlands, which occur in many regions around the world and provide a large number of important ecosystem services. We hope these results will trigger further research on these systems, including a more theoretical testing of the impermeable bedrock paradigm using a multiple model approach as suggested by the referee.

Response to Comment 2:

Calibration and validation periods covering a total duration of 3 years (and often much less) are the norm rather than the exception in physically-based hydrological modelling studies of wetlands (e.g. Ala-aho et al., 2017; Armandine Les Landes et al., 2014; Haahti et al., 2016; House et al., 2016; Levison et al., 2014; Li et al., 2019; Quillet et al., 2017; Thompson et al., 2004). In many cases this is the result of the unfortunate exclusion of wetland environments from formal hydrometric networks (e.g. Hollis and Thompson, 1998) despite their ecological and socio-economic significance. In revising the paper we will more explicitly refer to the often relatively limited calibration datasets available for wetland environments and that in comparison to many previous studies the data employed in the current study is both more numerous and spatially distributed

(e.g. Ala-aho et al., 2017; Armandine Les Landes et al., 2014; Haahti et al., 2016; Hammersmark et al., 2008; House et al., 2016; Li et al., 2019). The excellent fit of simulated long-term mean groundwater table depth, groundwater upwelling rate and groundwater seepage rate with observed mire boundaries (including along the narrow valley downstream and in the small sub-basins located upstream of and 30m above the main mire extent where no groundwater table depth data were available for calibration) demonstrates that the model satisfactorily reproduces the dominant hydrological characteristics of the mire and its catchment. We therefore conclude the model is suitably robust to quantify the long-term water balance of the mire. Like every model and for practical reasons, our model makes simplifying assumptions, one of which is uniform hydraulic conductivity throughout the fissured zone. We discuss the limitations associated with these simplifying assumptions, but demonstrate that they do not undermine the ability of the model to reproduce the dominant hydrological characteristics of the mire and its catchment and they do not undermine our conclusions.

Response to Comment 3:

We agree with Referee #1 and so when revising the paper will make it clearer that our sensitivity analysis is a local sensitivity analysis around the calibrated parameter values.

Response to Details:

p. 4, line 22: to date, there is still only limited coverage and availability of LiDAR data in France, and none was available for our site. We had no access to UAV equipment. Photogrammetry using the available aerial pictures would have resulted in lower accuracy due to their relatively coarse resolution (a state-of the-art photogrammetric workflow developed by the French Geographic Institute achieved residual mean planimetric and altimetric errors comprised between 1.0 and 3.5m and between 1.4 and 2.0m, respectively; Bris et al., 2018). We used the best technology and data that were available to us at the time to build the most accurate DEM that was possible.

p. 5, line 13: this is correct. This is described in depth in Duranel (2015) which is cited and available online (https://discovery.ucl.ac.uk/id/eprint/1472054/1/Duranel_PhDthesisADuranel2015.pdf) and, in response to a suggestion from Referee #2, to be expanded upon in the revised paper.

p. 7: we agree with Referee #1 and will update the revised paper accordingly.

p. 11, line 27: extensive attempts were made to tighten the convergence criteria, however this resulted in relatively small gains but a substantial increase in computing time. The water balance error is overwhelmingly caused by the MIKE SHE overland flow module within the mire, and so does not affect estimates of overland boundary inflow, stream inflow and groundwater inflow and outflow to and from the mire. Because the mire is saturated most of the time, very little infiltration is simulated within its boundaries. As a result, the overland flow component error mainly affects simulated overland outflow to the river, which is not the main focus of our study.

References cited in our response to Referee #1

Ala-aho, P., Soulsby, C., Wang, H. and Tetzlaff, D.: Integrated surface-subsurface model to investigate the role of groundwater in headwater catchment runoff generation: A minimalist approach to parameterisation, Journal of Hydrology, 547, 664–677, doi:10.1016/j.jhydrol.2017.02.023, 2017.

Armandine Les Landes, A., Aquilina, L., De Ridder, J., Longuevergne, L., Pagé, C. and Goderniaux, P.: Investigating the respective impacts of groundwater exploitation and climate change on wetland extension over 150 years, Journal of Hydrology, 509, 367–378, doi:10.1016/j.jhydrol.2013.11.039, 2014.

Bris, A. L., Giordano, S. and Mallet, C.: Vers une remise en géométrie automatique des prises de vues aériennes historiques photogrammétriques, Revue Française de Photogrammétrie et de Télédétection, 217–218, 11–23, 2018.

Duranel, A. J.: Hydrology and hydrological modelling of acidic mires in central France, PhD thesis, University College London, London, UK. [online] Available from: http://discovery.ucl.ac.uk/1472054/1/Duranel_PhDthesisADuranel2015.pdf, 2015.

Haahti, K., Warsta, L., Kokkonen, T., Younis, B. A. and Koivusalo, H.: Distributed hydrological modeling with channel network flow of a forestry drained peatland site, Water Resour. Res., 52(1), 246–263, doi:10.1002/2015WR018038, 2016.

Hammersmark, C. T., Rains, M. C. and Mount, J. F.: Quantifying the hydrological effects of stream restoration in a montane meadow, northern California, USA, River Res. Applic., 24(6), 735–753, doi:10.1002/rra.1077, 2008.

Hollis, G. E. and Thompson, J. R.: Hydrological data for wetland management, Water and Environment Journal, 12(1), 9–17, doi:10.1111/j.1747-6593.1998.tb00140.x, 1998.

House, A. R., Thompson, J. R., Sorensen, J. P. R., Roberts, C. and Acreman, M. C.: Modelling groundwater/surface water interaction in a managed riparian chalk valley wetland, Hydrol. Process., 30(3), 447–462, doi:10.1002/hyp.10625, 2016.

Levison, J., Larocque, M., Fournier, V., Gagné, S., Pellerin, S. and Ouellet, M. A.: Dynamics of a headwater system and peatland under current conditions and with climate change, Hydrol. Process., 28(17), 4808–4822, doi:10.1002/hyp.9978, 2014.

Li, Z., Gao, P. and Lu, H.: Dynamic changes of groundwater storage and flows in a disturbed alpine peatland under variable climatic conditions, Journal of Hydrology, 575, 557–568, doi:10.1016/j.jhydrol.2019.05.032, 2019.

Quillet, A., Larocque, M., Pellerin, S., Cloutier, V., Ferlatte, M., Paniconi, C. and Bourgault, M.-A.: The role of hydrogeological setting in two Canadian peatlands investigated through 2D steady-state groundwater flow modelling, Hydrological Sciences Journal, 62(15), 2541–2557, doi:10.1080/02626667.2017.1391387, 2017.

Thompson, J., Gavin, H., Refsgaard, A., Refstrup Sørenson, H. and Gowing, D.: Modelling the hydrological impacts of climate change on UK lowland wet grassland, Wetlands Ecology and Management, 17(5), 503–523, doi:10.1007/s11273-008-9127-1, 2009.

Thompson, J. R., Sørenson, H. R., Gavin, H. and Refsgaard, A.: Application of the coupled MIKE SHE/MIKE 11 modelling system to a lowland wet grassland in southeast England, Journal of Hydrology, 293(1–4), 151–179, doi:10.1016/j.jhydrol.2004.01.017, 2004.

———————————————————

---

## Author Comment (AC2) · 7 Aug 2020

We thank Anonymous Referee #2 for stating that our research question and case study are interesting and that our field measurement campaign that provided the data for model calibration and validation was comprehensive. We believe that we can significantly improve the paper by accounting for the referee's comments and suggestions.

General comment

Rather than testing a range of competing theoretical conceptualisations of our geological model as proposed by Referee #1 (a valid but entirely different approach), we chose

to use the most plausible conceptualisation based on the data available, while making reasonable simplifying assumptions to account for constraints associated with practical computing times and the risk of over-parameterisation. We fully agree with Referee #2 that a detailed description of how we built the conceptual model is required. An in-depth description is available in Duranel (2015) which is cited and is available online (https://discovery.ucl.ac.uk/id/eprint/1472054/1/Duranel_PhDthesisADuranel2015.pdf). Indeed, a whole chapter of this PhD thesis is devoted to the geological model which was built using a large dataset derived from multiple sources (existing unpublished data, data we collected as part of the project, and published literature and technical reports mainly in French).

In the original version of the paper we provided a concise description of the geology of the site and the geological model, considering that, since the geological model is described in the above publication, we should instead focus on the hydrological model and its results. Including more information on the geological model is certainly possible but will clearly make the paper a little longer. We believe that this will be necessary to respond to the referee's comments and would improve the paper. Therefore, we propose to revise Section 2.2 ("Geological model") of the paper to include 1) a new figure showing the two most informative Electrical Resistivity Tomography transects which were instrumental in defining the geological model; 2) as specifically requested by Referee #2, a new figure showing the results of previous drillings by the uranium mining company; and 3) a description and interpretation of the results obtained from the ERT survey and the geological drillings. The location of the two selected ERT transects and, as requested by Referee #2, the drillings, will be shown on an updated version of Figure 1 of the original version of the paper. We estimate that the proposed changes will add approximately two pages to the manuscript. Drafts of the proposed additional figures showing two selected ERT transects and the geological drillings are shown in Figure 1 and Figure 2 of this reply, respectively. Please note that their captions refer to Figure 1 of the original version of the paper.

[Figure]

Figure 1 of this reply shows the presence of conductive material underneath much more resistive material on the hilltops and slopes of the catchment. The transition between these layers is abrupt and its altitude is largest beneath hilltops. The depth of the transition is larger beneath steep slopes, and on all transects it intersects the surface topography in the valley precisely where the wetland boundary is located (taking into account the ERT positional accuracy). The only location where conductive material was recorded above resistive material is on one hilltop (visible on the left-hand side of transect a). This was interpreted as demonstrating that most of the material investigated corresponds to the densely fissured layer of a truncated granite weathering profile. The interface between resistive and conductive material was assumed to correspond to the groundwater table (the exfiltration of which led to the formation of the wetland). The increase in resistivity with depth at the very bottom of transect b) at a depth of around 55m below ground level was interpreted as a decrease in fissure density (and therefore bulk porosity), and a transition towards unweathered granite. The superficial conductive material on the hilltop was assumed to be (probably unsaturated) saprolite, which is more conductive than the fissured granite layer due to its larger clay and water contents. The configuration was not seen anywhere else within the study area, leading to the conclusion that on most hilltops and hillslopes the majority of saprolite has been eroded away and the combined thickness of the soil, periglacial deposits and remaining in-situ saprolite is too small (less than a metre) to be detected. There is no indication of the presence of a substantial saprolite layer in the valley bottom beneath the wetland, however the complete saturation of the profile may make the distinction between fissured granite and saprolite impossible in this area. Figure 2 of this reply shows that the granite is weathered to highly weathered, with dense fissuration leading to low or variable core recovery percentages, to a depth of 15-65m. A substantial saprolite layer ("grus") is present in five out of seven locations. This is seemingly at odds with the apparent absence of saprolite beneath the wetland on the ERT transects located further downstream. However it should be noted that the geological drillings are not representative of the entire catchment as they were undertaken within a small

area along uranium-rich mineralised faults of tectonic origin, which may have led to a more pronounced weathering locally. Our interpretation of the ERT and geological data agrees with the current understanding of granite weathering processes, granite landscape geomorphology and granite hydrogeology in the Massif Central in general and in the Monts d'Ambazac specifically (Desire-Marchand and Klein, 1986; Dewandel et al., 2006; Godard et al., 2001; Klein, 1978; Klein et al., 1990; Mauroux et al., 2009).

An important conclusion of the ERT survey is that the wetland appears to be hydrologically connected to, and most likely dependent on, a groundwater body within the fissured granite layer. Periglacial deposits and in-situ saprolite are patchy, and very shallow where present. Pedological pits on hillslopes and hilltops showed that soil texture is loamy-sandy to sandy-gravelly, and drainage is always good. Given these observations, we are of the opinion that, as raised by Referee #1, the "possibility that within the same layer parameters for the whole catchment there is combination of soil layers with regionally higher k and granite fissures with lower k" (which would imply that the soil layer plays a much larger role in lateral flow towards the wetland) is relatively small.

We developed our conceptual geological model taking into account the insights described above, and made the simplifying assumptions that:

- saturated flow occurs mostly in the fissured granite zone and in the peat;

- soil, saprolite and periglacial deposits can be neglected as far as saturated flow is concerned (but note that they are accounted for when modelling unsaturated flow);

- the hydrologically active fissured granite zone is 55m deep, follows the surface topography, and has homogeneous properties throughout.

Our computational model reflects these assumptions. Referee #2 is correct in stating that "the saturated zone of the model comprises of two computational layers. At peatlands the layer was 1) the peat as top layer and 2) fissured granite below; at mineral

soils 3) 2 meter thick layer on top and 4) fissured granite below. [...] all of the layer structures 2-4 had the same parameters (e.g. hydraulic conductivity). Given that there were some soil on top of the granite at least locally, all of the geological information outside the peat layer are within the parameters of layers 2-4." This indeed means soils (but also periglacial formations and the saprolite layer) are, for the reasons explained above, neglected as far as saturated flow is concerned. In revising the paper, we will make this important information much clearer justifying the assumptions with reference to the available geological information. We will also discuss more clearly the possible uncertainties considering the model structure as recommended by Referee #2.

Whilst we acknowledge that our conceptual and computational models, like any scientific model, are simplifications of reality, our results show that they are sufficiently accurate to reproduce observed water levels in a large number of piezometers, observed stream discharge at four locations distributed throughout the catchment upstream and downstream of the mire, and, importantly, the observed distribution of mire habitats (including along the narrow valley downstream and in the small sub-basins located upstream of and 30m above the main mire extent where no groundwater table depth data were available for calibration). We therefore conclude that they are appropriately accurate to reproduce the dominant hydrological characteristics of the mire and its catchment and to quantify its long-term water balance.

Specific comments:

"Is figure 9 needed in addition to figure 8?" Whilst Figure 8 gives the water balance terms in absolute terms, Figure 9 provides the proportional contribution of the three main sources of water to the mire. It highlights that, in proportion, the contribution of groundwater increases during the driest months. We feel this finding is important to better understand the eco-hydrology of this type of mire. This feature of our results cannot be as easily seen from Figure 8, which shows that, in absolute terms, groundwater inflow decreases during the driest months. Nevertheless, in response to this comment from Referee #2, we propose to move Figure 9 to the Supplement when revising the

paper thereby still demonstrating this finding but avoiding the potential repetition of similar figures.

Figures 10 and 11. We believe that Figure 11 is interesting in showing the progressive decline in the seepage rate from winter to late summer, and the almost perfect agreement between areas where seepage still occurs in September and the mire extent. We do, however, also acknowledge that the figure takes a large amount of space. Therefore we propose to move it to the Supplement. In revising the manuscript, we will follow the recommendations of Referee #2 and produce a new figure replacing Figures 10 and 11 in the main paper. As recommended, this figure will show the mean groundwater table depth, and the mean seepage rates in the driest and wettest months. Figure 3 of this reply provides a draft of the proposed new figure.

"Figure 12 is not opening up easily for the reader." We do believe that it is important to provide the time-series of hydrological data describing the wetland hydrology and used in the following analysis, in particular to demonstrate average hydrological condition in the wetland and to provide opportunities for comparison with other wetland systems. However, we acknowledge that we could provide some more discussion of this figure when it is first introduced. As recommended by Referee #2, in the revised paper, we propose to add vertical dashed lines marking a few points of interest on Figure 12, and to discuss these in the text. These lines will highlight the unusually low and high summer groundwater tables in 2005 and 2007, respectively. We will discuss these in view of the relatively high inter-annual variability of precipitation. We will also revise the background grid lines and improve their visibility to make it easier to relate patterns in the time-series to specific years.

Detailed comments:

"Page 4, line 29-30: Where are these boreholes on the map?" As described above, in revising the paper we will indicate the location of the boreholes on Figure 1.

"Page 16, line 20: Is the word evacuated the correct term?" In the revised paper we

will replace "they are quickly evacuated as saturation excess runoff" with "they quickly leave the wetland as saturation excess runoff".

"Figure 5. text. Explain what are the a-h standing for?" We are grateful to Referee #2 for pinpointing this missing text. The correct caption for Figure 5 should have been: "Observed mire boundary based on botanical (Durepaire and Guerbaa, 2008) and pedological (Duranel, 2015) criteria and the predicted mire distribution based on model results (simulated 2001–2013 mean groundwater table depth higher than 0.166 m below ground level). (a)-(h): refer to Section 3.1 for further explanation."

"Figure 13 text: explain abbreviations in x- and y-axes." Similarly, the correct caption for Figure 13 should have been: "Smooth terms of the final generalised additive model of simulated monthly mean groundwater table depth on the response scale. The curve shows the fitted value of the response, conditional upon the other explanatory variables being held at their sample mean. The shaded area is the approximate 95 % confidence interval. The time-series used are simulated spatially-averaged monthly means in the peat soil domain for the period 2001–2013: groundwater table depth (GWT), groundwater table depth in the preceding month (GWTm-1), upward saturated flow from the lower computational layer (SZ.up), actual evapotranspiration (ET) and precipitation (RR)." We apologise for these oversights, which will be corrected in the revised paper.

References cites in our reply to Referee #2

Desire-Marchand, J. and Klein, C.: Le relief du Limousin. Les avatars d'un géomorphotype, Norois, 129(1), 23–49, doi:10.3406/noroi.1986.4292, 1986.

Dewandel, B., Lachassagne, P., Wyns, R., Marechal, J. C. and Krishnamurthy, N. S.: A generalized 3-D geological and hydrogeological conceptual model of granite aquifers controlled by single or multiphase weathering, Journal of Hydrology, 330(1–2), 260–284, doi:10.1016/j.jhydrol.2006.03.026, 2006.

[Figure]

Duranel, A. J.: Hydrology and hydrological modelling of acidic mires in central France, PhD thesis, University College London, London, UK. [online] Available from: http://discovery.ucl.ac.uk/1472054/1/Duranel_PhDthesisADuranel2015.pdf, 2015.

Godard, A., Lagasquie, J. J. and Lageat, Y.: Basement Regions, Springer-Verlag, Berlin, Heidelberg, Germany., 2001.

Klein, C.: Les Monts d'Ambazac et de Saint-Goussaud. Deux points de vue sur la morphogenèse limousine, Norois, 97(1), 103–126, doi:10.3406/noroi.1978.3679, 1978.

Klein, C., Désiré-Marchand, J. and Giusti, C.: L'évolution géomorphologique de l'Europe hercynienne occidentale et centrale: aspects régionaux et essai de synthèse, Centre National de la Recherche Scientifique, Paris, France., 1990.

Mauroux, B., Wyns, R., Martelet, G. and Lions, J.: SILURES Limousin - Module 1 SILURES "Base de données". Recueil des données, interprétations et perspectives, Rapport final, Bureau de Recherches Géologiques et Minières, Orléans, France., 2009.
* * *
[Figure]

a)

Model resistivity with topography
Iteration 5 Abs. error = 5.3

SCH1264_1

Elevation

extent of pasture

wetland boundary
wetland boundary

Resistivity in ohm.m

300  600  1200  2400  4800  9600  19200  38400

Horizontal scale is 23.29 pixels per unit spacing
Vertical exaggeration in model section display = 0.89
First electrode is located at 0.0   m.
Last electrode is located at 395.0 m.

Unit Electrode Spacing = 5.00 m.

b)

Model resistivity with topography
Iteration 5 Abs. error = 2.2

SCH1264_1

Elevation

wetland boundary
stream
wetland boundary

Resistivity in ohm.m

200  398  791  1572  3125  6212  12350  24553

Horizontal scale is 12.77 pixels per unit spacing
Vertical exaggeration in model section display = 1.61
First electrode is located at 0.0   m.
Last electrode is located at 715.0 m.

Unit Electrode Spacing = 5.00 m.

**Fig. 1.** Selected Electrical Tomography Resistivity transects across the study site. The transect locations are shown on Figure 1.

**Borehole:** 1 2 3 4 5 6 7

**Legend**

**Weathering grade**
- grus
- highly weathered
- weathered
- slightly weathered
- mostly unweathered
- unweathered

**Core percentage**

- 0-10
- 10-20
- 20-30
- 30-40
- 40-50
- 50-60
- 60-70
- 70-80
- 80-90
- 90-100

**Fig. 2.** Geological drillings by the Commissariat à l'Energie Atomique, showing the granite weathering grade and the core recovery percentage. The drilling locations are shown on Figure 1.

[Figure]

**Fig. 3.** Simulated mean annual groundwater table depth and seepage rates in the wettest (January) and driest (September) months (2001–2013).

---

## Author Response (AR1)

**Modelling the hydrological interactions between a fissured granite aquifer and a valley mire in the Massif Central, France**

Arnaud Duranel[1,2], Julian R. Thompson[1], Helene Burningham[1], Philippe Durepaire[3], Stéphane Garambois[4], Robert Wyns[5], Hervé Cubizolle[2]

[1] UCL Department of Geography, University College London, London WC1E 6BT, United Kingdom
[2] Lyon University, UMR 5600 CNRS EVS, 42023 Saint-Etienne cedex 2, France
[3] Conservatoire d'Espaces Naturels de Nouvelle-Aquitaine, Réserve Naturelle Nationale de la Tourbière des Dauges, Sauvagnac, 87340 Saint-Léger-la-Montagne, France
[4] Université Grenoble Alpes, Univ. Savoie Mont Blanc, CNRS, IRD, IFSTTAR, ISTerre, UMR 5275, 38041 Grenoble, France
[5] Bureau de Recherches Géologiques et Minières, ISTO, UMR 7327, 45060 Orléans, France

Correspondence to: Arnaud Duranel (arnaud.duranel@univ-st-etienne.fr).

**RESPONSE TO REFEREES' COMMENTS**

**REFEREE #1**

We thank Anonymous Referee #1 for their comments and suggestions on the original paper. We believe that the paper has been improved considerably by responding to the issues that they raised and by incorporating their suggested revisions.

**Summary and Comment 1:**

The referee suggested that we "see the main generic contribution of the study as a demonstration of the limits of the "impermeable bedrock paradigm"". Although our study does indeed contradict this paradigm, we do not see this as its main contribution, nor its main objective. The "impermeable bedrock paradigm" was only referenced in the Introduction and Conclusion of the original paper, to frame our study within the wider context of hydrological research within hard rock regions. It is not central to the aims of the study and the methods employed which, as discussed below, focus on mire hydrology. Nevertheless, we do acknowledge that issues related to hard-rock hydrogeology figured prominently in the introduction of the initial manuscript. We felt this was necessary to set the scene for our research questions and highlight the novelty of our findings; but we appreciate this may have given the reader the impression that our study focuses on this issue. Therefore, in revising the manuscript, we have recast the Introduction to focus more explicitly on the central issues related to the hydrology of the type of wetland considered in our study. Given this, reference to the impermeable bedrock paradigm has been maintained but the central focus – i.e. modelling the hydrology of wetlands in such areas – is much more explicit.

The referee suggested that wetland processes could have been simulated with a multiple model approach, using different and competing conceptualizations of hydrological processes. While we fully agree that such an approach would have been

appropriate if our objectives had been to test the "impermeable bedrock paradigm" or to instead investigate the implications and associated uncertainty of using different process descriptions and associated data (e.g. Ruman et al., in press; Thompson et al., 2009, 2004), we argue the referee proposed a completely different study to that which the paper reports. Instead, our objectives, which are described at the end of the Introduction, clearly focus on the hydrology of the acidic valley mire. Using

5 a case study, we aimed to (i) test the ability of an equivalent porous medium approach, with limited data on the hydrodynamic properties of the granite weathering formations, to reproduce high-resolution spatial and temporal patterns in groundwater seepage and groundwater table depth within the mire, (ii) quantify the mire water balance including its dependence on groundwater inflows from granite weathering formations, and (iii) investigate the hydrological processes driving groundwater table depth in the mire. To do this we used the best available conceptual understanding of the research catchment and its mire,

10 supported by a large set of field data. In revising the manuscript, we have recast the Introduction and substantially expanded the Methods section to make our objectives and our methodological approach more explicit.

Whilst our research is a case study and uses established methods, to our knowledge this is the first integrated hydrological modelling study of a mire that accounts for groundwater flow in weathered hard rocks, and shows a close match between

15 spatial patterns of simulated groundwater upwelling and seepage and the observed distribution of mire habitats. As such, our results are novel and are of interest for a broad international readership since they have important implications for the hydrological understanding, management and conservation of such wetlands, which occur in many regions around the world (Cubizolle, 2019; Etlicher, 2005; Tanneberger et al., 2017) and provide a large number of important ecosystem services (Frolking et al., 2011; Okruszko et al., 2011; Parish et al., 2008; Yu et al., 2011). We believe these results will trigger further

20 research on these systems, including a more theoretical testing of the impermeable bedrock paradigm using a multiple model approach as suggested by the referee.

**Response to Comment 2:**

In revising the manuscript, we added a paragraph in Section 4.2 which explicitly discuss the often relatively limited calibration datasets available for wetland environments and that, in comparison to many previous studies (e.g. Ala-aho et al., 2017;

25 Armandine Les Landes et al., 2014; Haahti et al., 2016; Hammersmark et al., 2008; House et al., 2016; Li et al., 2019), the data employed in the current study is both more numerous and spatially distributed. The excellent fit of simulated long-term mean groundwater table depth, groundwater upwelling rate and groundwater seepage rate with observed mire boundaries (including along the narrow valley downstream and in the small sub-basins located upstream of and 30m above the main mire extent where no groundwater table depth data were available for calibration) demonstrates that the model reproduces the

30 dominant hydrological characteristics of the mire and its catchment. We therefore conclude the model is suitably robust to quantify the long-term water balance of the mire. Like every model and for practical reasons, our model makes simplifying assumptions, one of which is uniform hydraulic conductivity throughout the fissured zone. We discuss the limitations associated with these simplifying assumptions, but demonstrate that they do not undermine the ability of the model to reproduce

the dominant hydrological characteristics of the mire and its catchment and that they do not therefore undermine our conclusions.

**Response to Comment 3:**

We revised Sections 2.6 and 3.2 to make it clearer that our sensitivity analysis is a local sensitivity analysis around the calibrated parameter values.

**Response to Details:**

p. 4, line 22: To date, there is still only limited coverage and availability of LiDAR data in France, and none was available for our site. We had no access to UAV equipment. Photogrammetry using the available aerial photography would have resulted in lower accuracy due to their relatively coarse resolution (a state-of the-art photogrammetric workflow developed by the French Geographic Institute achieved residual mean planimetric and altimetric errors comprised between 1.0 and 3.5m and between 1.4 and 2.0m, respectively; Bris et al., 2018). We used the best technology and data that were available to us at the time to build the most accurate DEM that was possible.

p. 5, line 13: This is correct. In the revised manuscript and in response to a suggestion from Referee #2, we have substantially expanded the 'Geological model' part of our Methods section, to include a detailed description of the ERT survey results and interpretation.

p. 7: The revised manuscript now includes a paragraph in Section 2.3 ('Geological model') introducing the terms 'acrotelm' and 'catotelm'.

p. 11, line 27: In Section 3.3 of the revised manuscript, we added a paragraph explaining that (i) extensive attempts were made to tighten the convergence criteria, (ii) this resulted in relatively small gains but a substantial increase in computing time, (iii) and this is a relatively minor issue because it mainly affects simulated overland outflow to the river, which is not the main focus of our study.

**REFEREE #2**

**General comment**

We thank Anonymous Referee #2 for stating that our research question and case study are interesting and that our field measurement campaign that provided the data for model calibration and validation was comprehensive (thereby substantiating our comments made in response to Referee 1, comment 2). We believe that the paper has been significantly improved by accounting for the referee's comments and suggestions.

Rather than testing a range of competing theoretical conceptualisations of our geological model as proposed by Referee #1 (a valid but entirely different approach), we chose to use the most plausible conceptualisation based on the data available, while making reasonable simplifying assumptions to account for constraints associated with practical computing times and the risk

5    of over-parameterisation. We fully agree with Referee #2 that a detailed description of how we built the conceptual geological model and corresponding computational saturated zone model is therefore required. In response to their suggestions, we have significantly expanded the 'Geological model' part of the Methods section of the revised manuscript to include (i) a new figure (Figure 2) showing the two most informative Electrical Resistivity Tomography transects which were instrumental in defining the geological model; (ii) as specifically requested by Referee #2, a new figure (Figure 3) showing the results of previous

10    drillings by the uranium mining company; and (iii) a detailed description of the results obtained from the ERT survey and the geological drillings, and of their interpretation. We updated Figure 1 to show the location of the two selected ERT transects and, as requested by Referee #2, the drillings. An important conclusion of the ERT survey is that the wetland appears to be hydrologically connected to, and most likely dependent on, a groundwater body within the fissured granite layer. Periglacial deposits and in-situ saprolite are patchy, and very shallow where present. Pedological pits on hillslopes and hilltops showed

15    that soil texture is loamy-sandy to sandy-gravelly, and drainage is always good. Given these observations, we are of the opinion that, as raised by Referee #2, the "possibility that within the same layer parameters for the whole catchment there is combination of soil layers with regionally higher k and granite fissures with lower k" (which would imply that the soil layer plays a much larger role in lateral flow towards the wetland) is relatively small. When developing our computational saturated zone model, we made a number of simplifying assumptions. In response to Referee #2's comments, we have expanded Section

20    2.4 ('Hydrological model development') to make these assumptions much more explicit. We have also added a new table (Table 1) justifying each of these assumptions.

Whilst we acknowledge that our conceptual and computational models are, like any scientific model, simplifications of reality, our results show that they are sufficiently accurate to reproduce observed water levels in a large number of piezometers,

25    observed stream discharge at four locations distributed throughout the catchment upstream and downstream of the mire, and, importantly, the observed distribution of mire habitats (including along the narrow valley downstream and in the small sub-basins located upstream of and 30m above the main mire extent where no groundwater table depth data were available for calibration). We therefore conclude that they are appropriately accurate to reproduce the dominant hydrological characteristics of the mire and its catchment and to quantify its long-term water balance.

30    **Specific comments:**

*Is figure 9 needed in addition to figure 8?* Whilst Figure 8 of the initial manuscript gives the water balance terms in absolute terms, Figure 9 provides the proportional contribution of the three main sources of water to the mire. It highlights that, in proportion, the contribution of groundwater increases during the driest months. We believe that this finding is important to

better understand the eco-hydrology of this type of mire. This feature of our results cannot be as easily seen from Figure 8, which shows that, in absolute terms, groundwater inflow decreases during the driest months. Nevertheless, in response to this comment from Referee #2, we have moved Figure 9 to the Supplement (now Figure S3) in the revised paper thereby still demonstrating this finding but avoiding the potential repetition of similar figures. The text associated with the description of these results has accordingly been adjusted slightly to reflect this modification.

*Figures 10 and 11.* We believe that Figure 11 of the initial manuscript is interesting in showing the progressive decline in the seepage rate from winter to late summer, and the almost perfect agreement between areas where seepage still occurs in September and the mire extent based on the observed distribution of mire vegetation and soils. We do, however, also acknowledge that the figure takes a large amount of space. Therefore, when revising the paper we have moved it to the Supplement (now Figure S4). We also followed the recommendations of Referee #2 and produced a new figure (Figure 11) replacing figures 10 and 11 in the initial manuscript. As recommended, this figure shows the mean groundwater table depth, and the mean seepage rates in the driest and wettest months.

*Figure 12 is not opening up easily for the reader.* We do believe that it is important to provide the time-series of hydrological data describing the wetland's hydrology which is used in the analysis that follows, in particular to demonstrate average hydrological condition in the wetland and to provide opportunities for comparison with other wetland systems. To improve the readability of Figure 12, and as recommended by Referee #2, we have added vertical bands highlighting two contrasting summers, and a new paragraph discussing these in Section 3.5. We have also revised the background grid lines in Figure 12 to improve their visibility and make it easier to relate annual patterns in the time-series to specific years.

**Detailed comments:**

*Page 4, line 29-30. Where are these boreholes on the map?* We have revised Figure 1 to show the location of the boreholes.

*Page 16, line 20: Is the word evacuated the correct term?* We have replaced the original "they are quickly evacuated as saturation excess runoff" with "they quickly leave the wetland as saturated excess runoff".

*Figure 5. text. Explain what are the a-h standing for?* We have updated the caption of this figure (now Figure 7) to explain the meaning of these labels.

*Figure 13 text: explain abbreviations in x- and y-axes:* We have updated the caption of this figure to explain these abbreviations.

**References cited in our response to Referees.**

Ala-aho, P., Soulsby, C., Wang, H. and Tetzlaff, D.: Integrated surface-subsurface model to investigate the role of groundwater in headwater catchment runoff generation: A minimalist approach to parameterisation, J. Hydrol., 547, 664–677, doi:10.1016/j.jhydrol.2017.02.023, 2017.

5  Armandine Les Landes, A., Aquilina, L., De Ridder, J., Longuevergne, L., Pagé, C. and Goderniaux, P.: Investigating the respective impacts of groundwater exploitation and climate change on wetland extension over 150 years, J. Hydrol., 509, 367–378, doi:10.1016/j.jhydrol.2013.11.039, 2014.

Bris, A. L., Giordano, S. and Mallet, C.: Vers une remise en géométrie automatique des prises de vues aériennes historiques photogrammétriques, Rev. Fr. Photogrammétrie Télédétection, 217–218, 11–23, 2018.

10  Cubizolle, H.: Les tourbières et la tourbe, Lavoisier-Tec & Doc, Paris, France, 2019.

Etlicher, B.: French and Belgian Uplands, in The physical geography of Western Europe, vol. 6, edited by E. A. Koster, pp. 231–250, Oxford University Press, Oxford, UK, 2005.

Frolking, S., Talbot, J., Jones, M. C., Treat, C. C., Kauffman, J. B., Tuittila, E.-S. and Roulet, N.: Peatlands in the Earth's 21st century climate system, Environ. Rev., 19(NA), 371–396, doi:10.1139/a11-014, 2011.

15  Haahti, K., Warsta, L., Kokkonen, T., Younis, B. A. and Koivusalo, H.: Distributed hydrological modeling with channel network flow of a forestry drained peatland site, Water Resour. Res., 52(1), 246–263, doi:10.1002/2015WR018038, 2016.

Hammersmark, C. T., Rains, M. C. and Mount, J. F.: Quantifying the hydrological effects of stream restoration in a montane meadow, northern California, USA, River Res. Appl., 24(6), 735–753, doi:10.1002/rra.1077, 2008.

House, A. R., Thompson, J. R., Sorensen, J. P. R., Roberts, C. and Acreman, M. C.: Modelling groundwater/surface water
20  interaction in a managed riparian chalk valley wetland, Hydrol. Process., 30(3), 447–462, doi:10.1002/hyp.10625, 2016.

Li, Z., Gao, P. and Lu, H.: Dynamic changes of groundwater storage and flows in a disturbed alpine peatland under variable climatic conditions, J. Hydrol., 575, 557–568, doi:10.1016/j.jhydrol.2019.05.032, 2019.

Okruszko, T., Duel, H., Acreman, M., Grygoruk, M., Flörke, M. and Schneider, C.: Broad-scale ecosystem services of European wetlands—overview of the current situation and future perspectives under different climate and water management
25  scenarios, Hydrol. Sci. J., 56(8), 1501–1517, doi:10.1080/02626667.2011.631188, 2011.

Parish, F., Sirin, A., Charman, D., Joosten, H., Minayeva, T., Silvius, M. and Stringer, L.: Assessment on peatlands, biodiversity and climate change, Main report, Global Environment Centre & Wetlands International, Kuala Lumpur, Malaysia & Wageningen, The Netherlands. [online] Available from: http://www.gecnet.info/index.cfm?&menuid=48 (Accessed 29 June 2010), 2008.

30  Ruman, S., Ball, T., Black, A. R. and Thompson, J. R.: Influence of alternative representations of land use and geology on distributed hydrological modelling results: Eddleston, Scotland, Hydrol. Sci. J., in press.

Tanneberger, F., Tegetmeyer, C., Busse, S., Barthelmes, A., Shumka, S., Moles Mariné, Jenderedjian, K., Steiner, G. M., Essl, F., Etzold, J., Mendes, C., Kozulin, A., Frankard, P., Milanović, Đ., Ganeva, A., Apostolova, I., Alegro, A., Delipetrou, P., Navrátilová, J., Risager, M., Leivits, A., Fosaa, A. M., Tuominen, S., Muller, F., Bakuradze, T., Sommer, M., Christanis, K.,
35  Szurdoki, E., Oskarsson, H., Brink, S. H., Connolly, J., Bragazza, L., Martinelli, G., Aleksāns, O., Priede, A., Sungaila, D., Melovski, L., Belous, T., Saveljić, D., de Vries, F., Moen, A., Dembek, W., Mateus, J., Hanganu, J., Sirin, A., Markina, A.,

Napreenko, M., Lazarević, P., Šefferová Stanová, V., Skoberne, P., Heras Pérez, P., Pontevedra-Pombal, X., Lonnstad, J., Küchler, M., Wüst-Galley, C., Kirca, S., Mykytiuk, O., Lindsay, R. and Joosten, H.: The peatland map of Europe, Mires Peat, (19), 1–17, doi:10.19189/MaP.2016.OMB.264, 2017.

Thompson, J., Gavin, H., Refsgaard, A., Refstrup Sørenson, H. and Gowing, D.: Modelling the hydrological impacts of climate change on UK lowland wet grassland, Wetl. Ecol. Manag., 17(5), 503–523, doi:10.1007/s11273-008-9127-1, 2009.

Thompson, J. R., Sørenson, H. R., Gavin, H. and Refsgaard, A.: Application of the coupled MIKE SHE/MIKE 11 modelling system to a lowland wet grassland in southeast England, J. Hydrol., 293(1–4), 151–179, doi:10.1016/j.jhydrol.2004.01.017, 2004.

Yu, Z., Beilman, D. W., Frolking, S., MacDonald, G. M., Roulet, N. T., Camill, P. and Charman, D. J.: Peatlands and their role in the global carbon cycle, Eos Trans. Am. Geophys. Union, 92(12), 97–98, doi:10.1029/2011EO120001, 2011.

**LIST OF RELEVANT CHANGES**

**Abstract and Introduction**

The Abstract and Introduction have been recast to focus more explicitly on the central issues related to the hydrology of the type of wetland considered in our study.

5 **Methods**

The "Hydrometeorological monitoring" section has been moved forward to improve the manuscript's readability.

The "Geological model" section (now Section 2.3) has been significantly expanded to include a detailed description of the results obtained from the ERT survey and the geological drillings, and of their interpretation.

Section 2.3 now includes a paragraph introducing the terms 'acrotelm' and 'catotelm'.

10 The "Hydrological model development" section (Section 2.4) has been significantly recast and expanded to clarify and justify the assumptions we made when developing our saturated zone computational model.

The fact that our sensitivity analysis is a local sensitivity of the model around the calibrated parameter values has been made clearer in Sections 2.6 and 3.2.

**Results**

15 A paragraph has been added to Section 3.3 to further discuss the mire water balance computational error and demonstrate that it does not significantly impact our findings.

Sections 3.4 and 3.5 have been adjusted slightly to reflect changes in relevant figures, and to provide more discussion of Figure 12 when it is first introduced.

**Discussion**

20 A new paragraph has been added to Section 4.2 explicitly discussing the often relatively limited calibration datasets available for wetland environments and that, in comparison to many previous studies, the data employed in the current study is both more numerous and spatially distributed.

**References**

A small number of additional references have been included in line with changes in the main text.

25 **Figures**

We have updated Figure 1 to show the location of the two selected ERT transects and the drillings.

Two new figures have been added: Figure 2 shows the two most informative Electrical Resistivity Tomography transects, which were instrumental in defining the geological model, and Figure 3 shows the results of previous drillings by the uranium mining company.

The caption of Figure 7 (originally Figure 5) has been updated to explain the meaning of labels a-h.

The caption of Figure 10 (originally Figure 8) has been updated to explain the meaning of variable codes.

Figure 9 in the original manuscript has been moved to the Supplement (now Figure S3).

Figure 10 in the original manuscript (now Figure 11) has been modified to show mean groundwater seepage rates in January and September instead of mean annual groundwater seepage and upwelling rates.

Figure 11 in the original manuscript has been moved to the Supplement (now Figure S4).

Figure 12 now includes vertical bands highlighting two contrasting summers, further discussed in Section 3.5. We have also revised the background grid lines to improve their visibility and make it easier to relate annual patterns in the time-series to specific years.

The caption of Figure 13 has been updated to explain abbreviations used in the figure.

**Throughout the manuscript**

We have replaced the original "they are quickly evacuated as saturation excess runoff" with "they quickly leave the wetland as saturated excess runoff".

We have made a number of small changes to improve the readability of the manuscript and correct typos.

**Supplement**

We have updated Table S1 to explain abbreviations used in the table.

We have slightly recast panels in Figure S2 to avoid it being split over three pages.

We have moved Figures 9 and 11 of the original main manuscript to the Supplement (now Figures S3 and S4).

**REVISED MANUSCRIPT WITH MARKED-UP CHANGES**

[revised manuscript text omitted]

Electrical resistivity tomography (ERT) was used to estimate the depth of granite weathering formations along three 380 to 868 m long transects across the mire and the lower hillslopes, and one 315 m long transect located on the southern hilltop. A detailed description of these profiles is given in Duranel (2015). Figure 2 shows the two most informative profiles, the locations
25  of which are shown on Figure 1. A Schlumberger configuration was used in both cases, with an electrode spacing of 5 m and a maximum penetration depth of 60 m below ground level. The inversion was performed using the Res2Dinv software (Loke, 2013; Loke and Barker, 1996) considering a L1-optimization norm respectively leading to a RMS error of 5.3 % for profile A and 2.2 % for profile B. Both profiles are characterised by strong contrasts in resistivity values. The most striking feature is the presence on each side of the profiles of highly resistive material overlaying more conductive material. Highly resistive
30  material is only recorded beneath mineral soils on hilltops and slopes, and does not occur beneath the mire. The transition between these layers is very sharp, its altitude is largest beneath hilltops, and its depth greatest beneath steep slopes. Out of the four ERT profiles, the only location where conductive material was recorded above resistive material is on one hilltop (visible on the left-hand side of Profile A). These observations were interpreted as demonstrating that most of the material

investigated corresponds to the fissured layer of a truncated granite weathering profile. Highly resistive and conductive materials correspond to the unsaturated and saturated fissured layers, respectively. Taking into account the ERT positional accuracy, the margins of the mire coincided on all transects with the location where the inferred groundwater table reaches the ground surface, suggesting a determining role of groundwater seepage in the mire water balance and a high degree of hydrological connectivity between the mire and the fissured layer. The increase in resistivity with depth at the very bottom of Profile B at a depth of around 55 m below ground level was interpreted as a decrease in fissure density (and therefore bulk porosity), and a transition towards unweathered granite. The shallow (less than 2 m deep) superficial layer of conductive material on the hilltop on the left-hand side of Profile A is consistent with (probably unsaturated) saprolite, which is more conductive than the fissured granite layer due to its larger clay and water contents (Baltassat et al., 2005). This configuration was not seen anywhere else within the study area, leading to the conclusion that on most hilltops and hillslopes the majority of saprolite has been eroded away and the combined thickness of the soil, periglacial deposits and remaining in-situ saprolite is too small (less than a metre) to have been detected during the ERT survey. There is no indication of the presence of a substantial saprolite layer in the valley bottom beneath the mire, however the complete saturation of the profile may make the distinction between fissured granite and saprolite impossible in this area.

Large-scale underground uranium mining was undertaken in the area during the second part of the 20[th] century. Logs of seven geological boreholes drilled in 1973 by the Commissariat à l'Energie Atomique for the purpose of uranium exploration were provided by the mining company Areva (now Orano). These were all drilled along the eastern margin of the mire, across presumed mineralised faults located using ground surface radiological surveys. They were drilled at an angle of 49.5° relative to a horizontal plane to depths of between 67 m and 173 m. We used the degree of weathering (assessed on a six-class scale) and the core recovery percentage recorded at the time (Figure 3~~Electrical resistivity tomography (ERT) was used to estimate the depth of granite weathering formations along three 380 to 868m long transects located further east across the wetland and the lower hillslopes, and one 315 m long transect located on the southern hilltop. A combination of Schlumberger and Wenner arrays were used, with an electrode spacing of 5m and a maximum penetration depth of 60m below ground (Duranel, 2015).~~) to estimate the depth of the saprolite and of the fissured layer at those locations. The superficial 'grus' layer corresponds to saprolite, whereas granite recorded as 'weathered' to 'highly weathered', with dense fissuring leading to low or variable core recovery percentages, corresponds to the fissured layer. In line with ERT observations, the depth of the fissured layer ranges from 15 to 65 m, and its thickness from 38 to 65 m, with an average of 54 m. Again in line with ERT results, saprolite is absent from the upper part of the catchment (Borehole 7) and from steep slopes surrounding the mire (Borehole 1). However, a substantial saprolite layer 15–40 m deep is present in other boreholes located further inside the mire or on shallower slopes north-east and upstream of Puy Rond. This is seemingly at odds with the apparent absence of saprolite beneath the mire on ERT profiles, located downstream of Puy Rond. It should be noted that the geological boreholes are not representative of the entire catchment as they were drilled within a small area along uranium-rich mineralised faults of tectonic origin, which may have led to a deeper weathering front locally. Another explanation may be that the presence of Puy Rond, by creating a topographic bottleneck, has led to different saprolite erosion rates upstream and downstream of the hill.

Our interpretation of the ERT and geological data agrees with the current understanding of granite weathering processes, granite landscape geomorphology and granite hydrogeology in the Massif Central in general and in the Monts d'Ambazac specifically (Desire-Marchand and Klein, 1986; Dewandel et al., 2006; Godard et al., 2001; Klein, 1978; Klein et al., 1990; Mauroux et al., 2009). The absence of substantial saprolite formations from hilltops and slopes, and presumably from the downstream part of the valley, is in line with Mauroux et al. (2009), who used airborne spectral radiometry to demonstrate that saprolite has been largely eroded away in the Monts d'Ambazac.

[revised manuscript text omitted]

Saturated flow was modelled using an iterative implicit finite difference technique to solve the 3-dimensional Darcy equation including within the fissured granite for which an equivalent porous medium approach (Long et al., 1982) was used. In the development of the saturated flow model, we used the most plausible geological conceptualisation based on the data detailed in Sect. 2.3, while making a number of reasonable simplifying assumptions to account for constraints associated with practical

10    computing times, available data and the risk of over-parameterisation. These assumptions, further justified in Table 1, were (i) saturated flow occurs predominantly in the fissured granite layer and in the peat; (ii) mineral soils, saprolite, periglacial deposits and alluvium can be neglected as far as saturated flow is concerned; (iii) the hydrologically active fissured granite layer is 55m deep, follows the surface topography, and has homogeneous properties throughout; and (iv) peat deposits have homogeneous properties throughout.

15    Only two geological formations (peat and fissured granite) were therefore represented in the saturated flow model, with peat only occurring within the mire extent. The saturated flow model was nevertheless divided in two computational layers throughout the model extent, because MIKE SHE does not allow for discontinuous computational layers. Throughout the model, the lower layer represented the fissured granite, with a constant depth of 55 m below the topographic surface. Outside of the mire, the upper computational layer had identical properties to the lower layer and also represented the (upper) fissured

20    granite. In MIKE SHE the depth of the upper computational layer of the saturated flow component cannot be smaller than the depth of the root zone in the unsaturated flow component. Therefore, outside of the mire, the upper layer was defined as 2 m deep to match the estimated root depth of woodlands. Within the mire, where shallow-rooted open vegetation dominates, the upper layer represented peat deposits. Its depth followed that measured in the field, with a minimum value of 0.5 m to avoid numerical instabilities. 
[revised manuscript text omitted]

30  criteria, however this resulted in relatively small gains but a substantial increase in computing time. This is considered to be a minor issue because, as a result of the quasi-constant saturation, very little infiltration is simulated within the mire so. Consequently, the overland flow component error mainly affects simulated overland outflow to the river, which is not the main

focus of this study, and has little impact on other water balance terms. The water balance error within the mineral soils domain was only 1.6 % of total inflows. This is, again, mostly caused by the overland flow component . The three main sources of water to the mire are precipitation (32.1 % of total inflows), groundwater upwelling from the underlying mineral formations (27.1 %), and overland boundary inflow (i.e. non-channelized runoff from the mire catchment). The latter accounts for 40.2 % of total inflows, however a large proportion of this originates from edge-focused groundwater seepage on mineral soils just upstream of the mire (see Sect. 3.4). Lateral saturated boundary inflows from the mineral catchment and from the river are negligible (0.5 % and <0.0 % of total inflows respectively). Significantly, groundwater upwelling provides 92.4 % of inputs to the mire saturated zone, which drives groundwater levels. Percolation from the unsaturated zone to the saturated zone (5.3 %) and infiltration from overland to the saturated zone (0.7 %) are only possible when the groundwater table drops below ground level for 2–3 months in summer and early autumn. At all other times, inputs from precipitation and overland boundary flow quickly leave the mire as saturation-excess runoff and so do not contribute to the water balance of the saturated zone. Figure 10 breaks down the mire's long-term water balance by calendar month. The largest total inflows to the mire occur in winter and early spring. They then gradually decline throughout the spring and summer reaching a minimum in September. Total outflows, dominated by overland runoff to the stream and to a lesser extent by evapotranspiration, follow the same pattern. Changes in storage are relatively small, reflecting the very shallow groundwater levels and the rapid removal of any water surplus through saturation-excess runoff, particularly in late autumn, winter and early spring. Due to the site's position at the transition between a degraded oceanic climate and a mountainous climate (Joly et al., 2010), precipitation is, on average, relatively well distributed throughout the year although there is relatively large inter-annual variability. Overland boundary inflow is highly variable seasonally. It reaches its maximum in winter and early spring (accounting for 49.6 % of total inflows in January and February), before gradually declining to a minimum in September (22.9 %). It also exhibits large inter-annual variability reflecting the similar variability in precipitation. Upward saturated flow from the lower computational layer (i.e. groundwater upwelling) is relatively constant from December to May before gradually declining to reach a minimum in September. It is characterised by much smaller seasonal and inter-annual variabilities than precipitation and overland boundary inflow and, therefore, generally provides a larger proportion of total inflows in summer and early autumn (37.2 % in September, Figure S3 in the Supplement). As noted above, groundwater upwelling accounts for, on average, 92.4 % of total inflows to the mire saturated zone . Seasonally this figure peaks at 95.5 % in January and declines to 88.1 % in August. At this time receding groundwater levels allow for the formation of an unsaturated zone across a larger proportion of the mire and for precipitation and overland boundary inflow to infiltrate and percolate, respectively, instead of producing saturation-excess runoff.

**3.4 Spatial patterns in groundwater table depth,  and groundwater seepage**

Figure 11 shows the spatial distribution of simulated long-term (2001–2013) mean annual groundwater table depth and groundwater seepage rates in January and September when long-term mean seepage rates are

highest and lowest, respectively (data for other calendar months are shown in Figure S4 in the Supplement). Spatial and temporal patterns in groundwater upwelling rates are very similar to those in seepage rates and are therefore not shown. As discussed above, the mire boundaries are closely associated with shallow groundwater table depths. Groundwater upwelling and seepage rates show very similar spatial patterns and are highest along or immediately upstream of the mire boundary. This

5   is a clear example of edge-focused discharge (Richardson et al., 2001), whereby groundwater discharge at the ground surface is enhanced where there is a break in the slope of the surface topography and, consequently, of the water table gradient. The lower topographic slope within the mire itself leads to a rapid decrease in the vertical hydraulic gradient towards the centre of the mire, however the model suggests that groundwater seepage occurs throughout the mire. Localised discrepancies between observed mire boundaries and simulated groundwater table depth described in Sect. are also evident even in the spatial patterns

10   of groundwater upwelling and seepage, in particular along the south-eastern boundary of the mire.

shows the mean spatial distribution of simulated long term groundwater seepage for each calendardriest month (spatial patterns of groundwater upwelling are similar and are therefore not shown). . The extent of the area where groundwater seepage occurs peaks in winterJanuary, when it extends largely to mineral soils surrounding the mire and to the small unchannelized valleys and slope breaks in the upper part of the catchment. This area progressively shrinks, reaching its minimum extent in

15   September. It is at this time of the year that the agreement between spatial patterns in simulated groundwater seepage and observed mire boundaries is highest (kappa = 0.845). The monthly mean groundwater seepage threshold best discriminating between mire and non-mire vegetation during this calendar month is $0.005\ \mathrm{mm\ day^{-1}}$, suggesting that the mire boundaries are determined by the presence or absence of groundwater seepage during the driest month of the year. Nevertheless, minor localised discrepancies between observed mire boundaries and simulated groundwater table depth described in Sect. 3.1 are

20   also evident in the spatial patterns of groundwater upwelling and seepage, in particular along the south-eastern boundary of the mire.

**3.5 Processes driving groundwater table depth in peat soils**

Figure 12 shows time-series of spatially averaged monthly mean groundwater table depth and selected water balance items in the peat soils. Groundwater table depth shows low inter-annual variability in winter, being consistently within a few

25   centimetres of ground level. Inter-annual variability is larger in summer. The grey bands in Figure 12 highlight summers with the deepest (2005) and shallowest (2007) mean groundwater tables. These are concomitant with unusually low and high groundwater upwelling rates and overland boundary inflows, which themselves followed relatively dry and wet winters and springs, respectively. Evapotranspiration shows relatively little inter-annual variability, which can be explained by the relatively shallow groundwater table depth within the mire and the absence of water availability limitation. Groundwater

[revised manuscript text omitted]

**4.2 Modelling peatlands**

In comparison to some long-term hydrological modelling studies of larger catchments, the hydrological time-series assembled to calibrate and validate our model may appear relatively short. However, calibration and validation periods covering a total duration of 3 years (and often much less) are the norm rather than the exception in physically-based hydrological modelling studies of wetlands (e.g. Ala-aho et al., 2017; Armandine Les Landes et al., 2014; Haahti et al., 2016; House et al., 2016a; Levison et al., 2014; Li et al., 2019; Quillet et al., 2017; Thompson et al., 2004). In many cases this is the result of the unfortunate exclusion of wetland environments from formal hydrometric networks (e.g. Hollis and Thompson, 1998) despite their ecological and socio-economic significance.

[revised manuscript text omitted]

Armandine Les Landes, A., Aquilina, L., De Ridder, J., Longuevergne, L., Pagé, C. and Goderniaux, P.: Investigating the respective impacts of groundwater exploitation and climate change on wetland extension over 150 years, J. Hydrol., 509, 367–378, doi:10.1016/j.jhydrol.2013.11.039, 2014.

Baize, D. and Girard, M.-C., Eds.: Référentiel pédologique 2008, Association française pour l'étude du sol, Editions Quae, Versailles, France, 2009.

Baker, C., Thompson, J. R. and Simpson, M.: Hydrological dynamics I: surface waters, flood and sediment dynamics, in The Wetlands Handbook, edited by E. Maltby and T. Barker, pp. 120–168, Wiley-Blackwell, Chichester, UK, 2009.

5 Baltassat, J. M., Legchenko, A., Ambroise, B., Mathieu, F., Lachassagne, P., Wyns, R., Mercier, J. L. and Schott, J. J.: Magnetic resonance sounding (MRS) and resistivity characterisation of a mountain hard rock aquifer: the Ringelbach Catchment, Vosges Massif, France, Surf. Geophys., 3, 267–274, doi:10.3997/1873-0604.2005022, 2005.

Banks, E. W., Simmons, C. T., Love, A. J., Cranswick, R., Werner, A. D., Bestland, E. A., Wood, M. and Wilson, T.: Fractured bedrock and saprolite hydrogeologic controls on groundwater/surface-water interaction: a conceptual model (Australia), 10 Hydrogeol. J., 17(8), 1969–1989, doi:10.1007/s10040-009-0490-7, 2009.

Belyea, L. R. and Clymo, R. S.: Feedback control of the rate of peat formation, Proc. R. Soc. Lond. B Biol. Sci., 268(1473), 1315–1321, doi:10.1098/rspb.2001.1665, 2001.

Boelter, D. H.: Water storage characteristics of several peats in situ, Soil Sci. Soc. Am. J., 28(3), 433–435, doi:10.2136/sssaj1964.03615995002800030039x, 1964.

15 Boelter, D. H.: Important physical properties of peat materials, in Proceedings of the third international peat congress, pp. 150–154, Department of Energy, Mines and Resources and National Research Council of Canada, Quebec, Canada, 1968.

Boeye, D. and Verheyen, R. F.: The hydrological balance of a groundwater discharge fen, J. Hydrol., 137(1–4), 149–163, doi:10.1016/0022-1694(92)90053-X, 1992.

Bourgault, M. A., Larocque, M. and Roy, M.: Simulation of aquifer-peatland-river interactions under climate change, 20 Hydrol. Res., 45(3), 425–440, doi:10.2166/nh.2013.228, 2014.

Brandyk, T., Szatylowicz, J., Oleszczuk, R. and Gnatowski, T.: Water-related physical attributes of organic soils, in Organic soils and peat materials for sustainable agriculture, edited by L. E. Parent and P. Ilnicki, pp. 33–66, CRC Press, Boca Raton, Florida, USA, 2002.

Branfireun, B. A. and Roulet, N. T.: The baseflow and storm flow hydrology of a precambrian shield headwater peatland, 25 Hydrol. Process., 12, 57–72, doi:10.1002/(SICI)1099-1085(199801)12:1%3C57::AID-HYP560%3E3.0.CO;2-U, 1998.

Brown, P. A., Gill, S. A. and Allen, S. J.: Metal removal from wastewater using peat, Water Res., 34(16), 3907–3916, doi:10.1016/S0043-1354(00)00152-4, 2000.

Brunner, P. and Simmons, C. T.: HydroGeoSphere: a fully integrated, physically based hydrological model, Ground Water, 30 50(2), 170–176, doi:10.1111/j.1745-6584.2011.00882.x, 2012.

Calder, I. R.: Assessing the water use of short vegetation and forests: development of the Hydrological Land Use Change (HYLUC) model, Water Resour. Res., 39(11), 1318, doi:10.1029/2003WR002040, 2003.

Clilverd, H. M., Thompson, J. R., Heppell, C. M., Sayer, C. D. and Axmacher, J. C.: Coupled hydrological/hydraulic modelling of river restoration impacts and floodplain hydrodynamics, River Res. Appl., (32), 1927–1948, doi:10.1002/rra.3036, 35 2016.

Commission of the European Communities, Ed.: CORINE biotopes manual - A method to identify and describe consistently sites of major importance for nature conservation - Data specifications, Office for Official Publications of the European Communities, Luxembourg, 1991.

Congalton, R. G.: A review of assessing the accuracy of classifications of remotely sensed data, Remote
5  Sens. Environ., 37(1), 35–46, doi:10.1016/0034-4257(91)90048-B, 1991.

Cubizolle, H.: Les tourbières et la tourbe, Lavoisier-Tec & Doc, Paris, France, 2019.

Delgado, J., Llorens, P., Nord, G., Calder, I. R. and Gallart, F.: Modelling the hydrological response of a Mediterranean medium-sized headwater basin subject to land cover change: The Cardener River basin (NE Spain), J. Hydrol., 383(1–2), 125–134, doi:10.1016/j.jhydrol.2009.07.024, 2010.

10  Derrière, N., Wurpillot, S. and Vidal, C.: Un siècle d'expansion des forêts françaises - De la statistique Daubrée à l'inventaire forestier de l'IGN, L'If, 31, 1–8, 2013.

Desire-Marchand, J. and Klein, C.: Le relief du Limousin. Les avatars d'un géomorphotype, Norois, 129(1), 23–49, doi:10.3406/noroi.1986.4292, 1986.

[revised manuscript text omitted]

10  Maréchal, J. C., Wyns, R., Lachassagne, P. and Subrahmanyam, K.: Vertical anisotropy of hydraulic conductivity in the fissured layer of hard-rock aquifers due to the geological structure of weathering profiles, Journal of the Geological Society ofJ. Geol. Soc. India, 63(5), 545–550, 2004b2004.

Mauroux, B., Wyns, R., Martelet, G. and Lions, J.: SILURES Limousin - Module 1 SILURES "Base de données". Recueil des données, interprétations et perspectives, Rapport final, Bureau de Recherches Géologiques et Minières, Orléans, France, 2009.

15  Moriasi, D. N., Arnold, J. G., Van Liew, M. W., Bingner, R. L., Harmel, R. D. and Veith, T. L.: Model evaluation guidelines for systematic quantification of accuracy in watershed simulations, Transactions of the American Society of Agricultural and Biological Engineers,Trans. Am. Soc. Agric. Biol. Eng., 50(3), 885–900, doi:10.13031/2013.23153, 2007.

Morley, T. R., Reeve, A. S. and Calhoun, A. J. K.: The role of headwater wetlands in altering streamflow and chemistry in a Maine, USA catchment, JAWRA Journal of the AmericanJ. Am. Water Resources Association,Resour. Assoc., 47(2), 337–
20  349, doi:10.1111/j.1752-1688.2010.00519.x, 2011.

Muller, F.: Strategies for peatland conservation in France - a review of progress, Mires and Peat, (21), 1–13, doi:10.19189/MaP.2016.OMB.218, 2018.

Munger, J. L., Pellerin, S., Larocque, M. and Ferlatte, M.: Espèces végétales indicatrices des échanges d'eau entre tourbière et aquifère, Le Naturaliste canadien,Nat. Can., 138(1), 4–12, doi:10.7202/1021038ar, 2014.

25  Mustamo, P., Hyvärinen, M., Ronkanen, A.-K. and Kløve, B.: Physical properties of peat soils under different land use options, Soil Use Manage,Manag., 32(3), 400–410, doi:10.1111/sum.12272, 2016.

Neuman, S. P.: Trends, prospects and challenges in quantifying flow and transport through fractured rocks, Hydrogeol. J., 13(1), 124–147, doi:10.1007/s10040-004-0397-2, 2005.

Okruszko, T., Duel, H., Acreman, M., Grygoruk, M., Flörke, M. and Schneider, C.: Broad-scale ecosystem services of
30  European wetlands—overview of the current situation and future perspectives under different climate and water management scenarios, Hydrological Sciences Journal,Hydrol. Sci. J., 56(8), 1501–1517, doi:10.1080/02626667.2011.631188, 2011.

Parish, F., Sirin, A., Charman, D., Joosten, H., Minayeva, T., Silvius, M. and Stringer, L.: Assessment on peatlands, biodiversity and climate change, Main report, Global Environment Centre & Wetlands International, Kuala Lumpur, Malaysia & Wageningen, The Netherlands, 2008.

Quillet, A., Larocque, M., Pellerin, S., Cloutier, V., Ferlatte, M., Paniconi, C. and Bourgault, M.-A.: The role of hydrogeological setting in two Canadian peatlands investigated through 2D steady-state groundwater flow modelling, Hydrol. Sci. J., 62(15), 2541–2557, doi:10.1080/02626667.2017.1391387, 2017.

Reeve, A. S. and Gracz, M.: Simulating the hydrogeologic setting of peatlands in the Kenai Peninsula Lowlands, Alaska, Wetlands, 28(1), 92–106, doi:10.1672/07-71.1, 2008.

Refsgaard, J. C., Storm, B. and Clausen, T.: Système Hydrologique Europeén (SHE): review and perspectives after 30 years development in distributed physically-based hydrological modelling, Hydrol. Res., 41(5), 355, doi:10.2166/nh.2010.009, 2010.

Richardson, J. L., Arndt, J. L. and Montgomery, J. A.: Hydrology of wetland and related soils, in Wetland soils - Genesis, hydrology, landscapes, and classification, edited by J. L. Richardson and M. J. Vepraskas, pp. 35–84, CRC Press, Boca Raton, Florida, USA, 2001.

Rochester, R. E. L.: Uncertainty in hydrological modelling: a case study in the Tern catchment, Shropshire, UK, PhD thesis, University College London, London, UK, [online] Available from: http://discovery.ucl.ac.uk/795428/ (Accessed 24 November 2011), 2010.

Romanov, V. V.: Hydrophysics of Bogs [Gidrofizika bolot (1961)], edited by A. Heimann, Israel Program for Scientific Translation, Jerusalem, Israel, 1968.

Rossi, P. M., Ala-aho, P., Ronkanen, A.-K. and Kløve, B.: Groundwater–surface water interaction between an esker aquifer and a drained fen, J. Hydrol., 432–433, 52–60, doi:10.1016/j.jhydrol.2012.02.026, 2012.

Rydin, H. and Jeglum, J. K.: The biology of peatlands, Oxford University Press, Oxford, UK, 2006.

Šanda, M., Vitvar, T., Kulasová, A., Jankovec, J. and Císlerová, M.: Run-off formation in a humid, temperate headwater catchment using a combined hydrological, hydrochemical and isotopic approach (Jizera Mountains, Czech Republic), Hydrol. Process., 28(8), 3217–3229, doi:10.1002/hyp.9847, 2014.

van der Schaaf, S.: Analysis of the hydrology of raised bogs in the Irish Midlands. A case study of Raheenmore bog and Clara bog, Doctoral thesis, Wageningen Agricultural University, Wageningen, The Netherlands, 1999.

van der Schaaf, S.: Bog hydrology, in Conservation and restoration of raised bogs: geological, hydrological and ecological studies, edited by M. G. C. Schouten, pp. 54–109, Stationery Office, Dublin, Ireland, 2002.

Siegel, D. I. and Glaser, P. H.: Groundwater flow in a bog-fen complex, Lost River peatland, northern Minnesota, J. Ecol., 75(3), 743–754, doi:10.2307/2260203, 1987.

Singhal, B. B. S. and Gupta, R. P.: Applied hydrogeology of fractured rocks, 2nd ed., Springer, Dordrecht, The Netherlands, 2010.

Sobolewski, A.: A review of processes responsible for metal removal in wetlands treating contaminated mine drainage, Int. J. Phytoremediation, 1(1), 19–51, doi:10.1080/15226519908500003, 1999.

Tanneberger, F., Tegetmeyer, C., Busse, S., Barthelmes, A., Shumka, S., Moles Mariné, Jenderedjian, K., Steiner, G. M., Essl, F., Etzold, J., Mendes, C., Kozulin, A., Frankard, P., Milanović, Đ., Ganeva, A., Apostolova, I., Alegro, A., Delipetrou, P., Navrátilová, J., Risager, M., Leivits, A., Fosaa, A. M., Tuominen, S., Muller, F., Bakuradze, T., Sommer, M., Christanis, K., Szurdoki, E., Oskarsson, H., Brink, S. H., Connolly, J., Bragazza, L., Martinelli, G., Aleksāns, O., Priede, A., Sungaila, D.,

Melovski, L., Belous, T., Saveljić, D., de Vries, F., Moen, A., Dembek, W., Mateus, J., Hanganu, J., Sirin, A., Markina, A., Napreenko, M., Lazarević, P., Šefferová Stanová, V., Skoberne, P., Heras Pérez, P., Pontevedra-Pombal, X., Lonnstad, J., Küchler, M., Wüst-Galley, C., Kirca, S., Mykytiuk, O., Lindsay, R. and Joosten, H.: The peatland map of Europe, Mires Peat, (19), 1–17, doi:10.19189/MaP.2016.OMB.264, 2017.

5   Thompson, J., Gavin, H., Refsgaard, A., Refstrup Sørenson, H. and Gowing, D.: Modelling the hydrological impacts of climate change on UK lowland wet grassland, Wetl. Ecol. Manag., 17(5), 503–523, doi:10.1007/s11273-008-9127-1, 2009.

Thompson, J. R.: Simulation of wetland water-level manipulation using coupled hydrological/hydraulic modeling, Phys. Geogr., 25(1), 39–67, doi:10.2747/0272-3646.25.1.39, 2004.

Thompson, J. R.: Modelling the impacts of climate change on upland catchments in southwest Scotland using MIKE SHE and
10   the UKCP09 probabilistic projections, Hydrol. Res., 43(4), 507, doi:10.2166/nh.2012.105, 2012.

Thompson, J. R., Sørenson, H. R., Gavin, H. and Refsgaard, A.: Application of the coupled MIKE SHE/MIKE 11 modelling system to a lowland wet grassland in southeast England, J. Hydrol., 293(1–4), 151–179, doi:10.1016/j.jhydrol.2004.01.017, 2004.

Thompson, J. R., Iravani, H., Clilverd, H. M., Sayer, C. D., Heppell, C. M. and Axmacher, J. C.: Simulation of the hydrological
15   impacts of climate change on a restored floodplain, Hydrol. Sci. J., 62(15), 2482–2510, doi:10.1080/02626667.2017.1390316, 2017.

Tromp-van Meerveld, H. J., Peters, N. E. and McDonnell, J. J.: Effect of bedrock permeability on subsurface stormflow and the water balance of a trenched hillslope at the Panola Mountain Research Watershed, Georgia, USA, Hydrol. Process., 21(6), 750–769, doi:10.1002/hyp.6265, 2007.

20   Uchida, T., Asano, Y., Ohte, N. and Mizuyama, T.: Seepage area and rate of bedrock groundwater discharge at a granitic unchanneled hillslope, Water Resour. Res., 39(1), 1018, doi:10.1029/2002WR001298, 2003.

Valadas, B.: Les hautes terres du Massif Central français: contribution à l'étude des morphodynamiques récentes sur versants cristallins et volcaniques, Thèse de Doctorat d'Etat, Université Paris I, Paris, France, 1984.

Valadas, B.: L'alvéole des Dauges : un modèle géomorphologique, Ann. Sci. Limousin, N° spécial :
25   Tourbière des Dauges, 5–13, 1998.

Verger, J. P.: Les sols de l'alvéole de la tourbière du ruisseau des Dauges (Limousin), Ann. Sci. Limousin, N° spécial : Tourbière des Dauges, 43–54, 1998.

Vernoux, J. F., Llons, J., Petelet-Giraud, E., Seguin, J. J. and Stollsteiner, P.: Contribution à la caractérisation des relations entre eau souterraine, eau de surface et écosystèmes terrestres associés en lien avec la DCE, Rapport final, Bureau de
30   Recherches Géologiques et Minières, Orléans, France, 2010.

Wassen, M. J., Barendregt, A., Schot, P. P. and Beltman, B.: Dependency of local mesotrophic fens on a regional groundwater flow system in a poldered river plain in the Netherlands, Landsc. Ecol., 5(1), 21–38, doi:10.1007/BF00153801, 1990.

Wheeler, B. D., Gowing, D. J. G., Shaw, S. C., Mountford, J. O. and Money, R. P.: Ecohydrological guidelines for lowland
35   wetland plant communities, Environment Agency (Anglian Region), Peterborough, UK, 2004.

Whiteman, M., Brooks, A., Skinner, A. and Hulme, P.: Determining significant damage to groundwater-dependent terrestrial ecosystems in England and Wales for use in implementation of the Water Framework Directive, Ecol. Eng., 36(9), 1118–1125, doi:10.1016/j.ecoleng.2010.03.013, 2010.

Whitfield, P. H., St-Hilaire, A. and van der Kamp, G.: Improving hydrological predictions in peatlands, Can. Water Resour. J., 34(4), 467–478, doi:10.4296/cwrj3404467, 2009.

Wood, S.: Package 'mgcv'. Mixed GAM computation vehicle with automatic smoothness estimation. [online] Available from: http://cran.r-project.org/web/packages/mgcv, 2016.

Worrall, F., Chapman, P., Holden, J., Evans, C., Artz, R., Smith, P. and Grayson, R.: A review of current evidence on carbon fluxes and greenhouse gas emissions from UK peatlands, JNCC report, Joint Nature Conservation Committee, Peterborough, UK, 2011.

Wyns, R., Baltassat, J.-M., Lachassagne, P., Legchenko, A., Vairon, J. and Mathieu, F.: Application of proton magnetic resonance soundings to groundwater reserve mapping in weathered basement rocks (Brittany, France), Bull. Société Géologique Fr., 175(1), 21–34, doi:10.2113/175.1.21, 2004.

Yan, J. and Smith, K. R.: Simulation of integrated surface water and ground water systems-model formulation, J. Am. Water Resour. Assoc., 30(5), 879–890, doi:10.1111/j.1752-1688.1994.tb03336.x, 1994.

Yu, Z., Beilman, D. W., Frolking, S., MacDonald, G. M., Roulet, N. T., Camill, P. and Charman, D. J.: Peatlands and their role in the global carbon cycle, Eos Trans. Am. Geophys. Union, 92(12), 97–98, doi:10.1029/2011EO120001, 2011.

[Figure]

[Figure]

**Figure 1. The Dauges mire, its catchment and the hydrological monitoring network.**

[Figure]

**Figure 2.** **Electrical Resistivity Tomography images obtained after the inversion of apparent resistivity data acquired using a Schlumberger configuration. The survey was conducted using 80 electrodes for Profile A and 144 electrodes for Profile B, with an electrode spacing of 5 m. The inversion was performed using the Res2Dinv software considering a L1-optimization norm respectively leading to a RMS error of 5.3 % for Profile A and 2.2 % for Profile B. White dots show the centres of model blocks, the black line the surface topography and the black vertical marks the electrode positions. The profile locations are shown in Figure 1.**

[Figure]

**Figure 3. Geological boreholes drilled by the Commissariat à l'Energie Atomique, showing the granite weathering grade and the core recovery percentage. The borehole locations are shown in Figure 1. The 'highly weathered' and 'weathered' layers were interpreted as corresponding to the fissured layer, while the 'slightly weathered' to 'unweathered' layers were interpreted as corresponding to the bedrock.**

[Figure]

[Figure]

**Figure .** Water balance domains used within the MIKE SHE / MIKE 11 model of the Dauges catchment.

[Figure]

**Figure 5. Observed and simulated stream discharge and model performance statistics for two selected locations within the Dauges catchment (01/01/2011–31/12/2013). Note different y-axis ranges.**

[Figure]

5    **Figure 6. Observed and simulated groundwater table depth and model performance statistics for selected dipwells within the Dauges catchment (01/01/2011–31/12/2013). Note different y-axis ranges.**

[Figure]

 **Figure 6 (continued). Observed and simulated groundwater table depth and model performance statistics for selected dipwells within the Dauges catchment (01/01/2011–31/12/2013). Note different y-axis ranges.**

[Figure]

**Figure 7. Observed mire boundary based on botanical (Durepaire and Guerbaa, 2008) and pedological (Duranel, 2015) criteria and the predicted mire distribution based on model results (simulated 2001–2013 mean groundwater table depth higher than 0.166 m below ground level). (a)-(h) refer to locations discussed in the text (see Sect. 3.1).**

[Figure]

**Figure 8. Summary of model sensitivity analysis results (see Table 3 for parameter codes). Parameters are arranged in order of increasing mean sensitivity from bottom to top.**

[Figure]

**Figure 9. Simulated mean annual water balance of the Dauges mire (mm yr$^{-1}$) for the period 2001–2013. Evap.: evapotranspiration; interc.: interception; transp.: transpiration; infilt: infiltration; percol.: percolation; OL: overland; UZ: unsaturated zone; SZ: saturated zone.**

[Figure]

**Figure 10. Simulated mean monthly water balance of the Dauges mire (2001–2013). Figures follow the MIKE SHE water balance convention: inputs are negative, outputs positive, change in storage is positive when storage increases, and the water balance error is the sum of all inputs and outputs. Whiskers show the standard deviation. Top plot: see titles of bottom plots for variable codes.**

[Figure]

Figure 11

Figure . Mean monthly proportion of simulated inflow to the Dauges mire from different sources (2001–2013). Whiskers show the standard deviation. Lateral saturated boundary flow from the mineral catchment and from the river account for less than 0.6 % of total inflows at all times and are therefore not shown.

[Figure]

**Figure .** **Simulated mean annual groundwater table depth, upward and groundwater flow and seepage raterates in January and September (2001–2013). SZ: saturated zone; OL: overland.**

[Figure]

[Figure]

[Figure]

Figure . Simulated mean monthly groundwater seepage rate (2001–2013).

**Figure 12. Monthly means of simulated groundwater table depth and selected water balance items in the peat soil domain (2001–2013). Grey bands highlight two contrasting summers further described in Sect. 3.5.**

[Figure]

**Figure 13. Smooth terms of the final generalised additive model of simulated monthly mean groundwater table depth on the response scale. The curve shows the fitted value of the response, conditional upon the other explanatory variables being held at their sample mean. The shaded area is the approximate 95 % confidence interval. The time-series used are simulated spatially-averaged monthly means in the peat soil domain for the period 2001–2013: groundwater table depth (GWT), groundwater table depth in the preceding month (GWT$_{m-1}$), upward saturated flow from the lower computational layer (SZ.up), actual evapotranspiration (ET) and precipitation (RR).**

[Figure]

**Figure 14. Venn diagram of the variation partitioning results. The diagram shows the proportion of deviance in groundwater table depth (GWT) explained independently or jointly by groundwater table depth in the preceding month (GWT$_{m-1}$), upward saturated flow from the lower computational layer (SZ.up), actual evapotranspiration (ET) and/or precipitation (RR), based on simulated spatially-averaged monthly means in the peat soil domain for the period 2001–2013.**

**Table 1. Justification of simplifying assumptions of the saturated flow model component.**

| Simplifying assumption | Justification |
|---|---|
| Saturated flow occurs predominantly in the fissured granite layer and in the peat; mineral soils, saprolite, periglacial deposits and alluvium can be neglected as far as saturated flow is concerned | Except on one hilltop, ERT showed the combined thickness of mineral soils, periglacial deposits and saprolite to be too small (less than a metre) to be detected. Peri-glacial deposits are very patchy. Drainage in mineral soils is good. Alluvium was only found in a limited area along the stream downstream of Puy Rond.

 In MIKE SHE, the depth of the upper computational layer of the saturated flow component cannot be smaller than the depth of the root zone of the unsaturated flow component. Due to the shallowness of mineral soils, periglacial deposits and saprolite, and because most of mineral soils are covered with deep-rooted vegetation (deciduous woodland), there is a trade-off between adequately representing (i) saturated flow and (ii) unsaturated flow and evapotranspiration in these formations.

 ERT suggested the water table is located deep within the fissured layer on hilltops and hillslopes, and intersects the ground surface precisely at the mire boundary, suggesting a determining role of seepage of groundwater from the fissured zone in the mire water balance. |
| The hydrologically active fissured granite layer is 55m deep, follows the surface topography, and has homogeneous properties throughout | Geological boreholes showed the thickness of the fissured zone to range from 38 m to 65 m, with an average of 54 m. This is consistent with ERT observations, which showed an increase in resistivity at a depth of around 55 m below ground level (Transect B in Figure 2). Similar values were recorded in Borehole 7, located at a higher altitude, suggesting the fissured zone broadly follows the surface topography.

 Hydrodynamic properties of the fissured zone could not be measured and had to be estimated through model calibration. An assumption of homogeneous properties reduces computing time and the risk of over-parameterisation. |
| Peat deposits have homogeneous properties throughout | Slug tests and stratigraphical surveys showed the acrotelm to be less than 25 cm deep throughout most of the mire extent, which is too shallow for the acrotelm to be modelled independently in MIKE SHE without numerical instabilities and without impacting unsaturated flow and evapotranspiration modelling. |

[revised manuscript text omitted]